

# A lightweight balloon-borne mid-infrared hygrometer to probe the middle atmosphere: Pico-Light H₂O. Comparison with Aura-MLS v4 and v5 satellite measurements

Mélanie Ghysels[1], Georges Durry[1], Nadir Amarouche[2], Jean-Christophe Samake[2], Fabien Frérot[2], Emmanuel. D. Rivière[1]

[1]Groupe de Spectrométrie Moléculaire et Atmosphérique (GSMA, UMR CNRS 7331), Université de Reims, UFR Sciences Exactes et Naturelles, Moulin de la Housse B.P. 1039, 51687 Reims Cedex 2,FRANCE
[2]INSU Division Technique, 1 place Aristide Briand, 92195 Meudon cedex, FRANCE

*Correspondence to*: Mélanie Ghysels (melanie.ghysels-dubois@univ-reims.fr)

**Abstract.** Newly developed mid-infrared lightweight hygrometer, Pico-Light H₂O has been tested in-flight on February 19, 2019 and October 16, 2019. It has been flown under a 1200g rubber balloon operated by CNES from the Aire-sur-l'Adour facility (France) within the E.U. funded HEMERA WP11. During these two flights, we were able to obtain coincident MLS v4 and v5 water vapor and temperature profiles, leading to an inter-comparison between Pico-Light and Aura-MLS water vapor and temperature retrievals. Results from the comparison are in line with previous reported studies . Here, differences in the mid-latitude stratosphere and upper troposphere (20-316 hPa) are within 7% and 64% respectively. Largest differences with MLS v4 occurring within the upper troposphere nearby the cold point tropopause. The v5 MLS data have been corrected for observed dry bias nearby the tropopause, allowing to partially solve the observed discrepancies. Additionally, on February 19, the hygrometer has flown within an air filament from polar latitudes most of the flight for which a signature is observed on the water vapor profile and confirmed with ozone reanalysis from ERA 5and potential vorticity from MIMOSA advection model .

## 1. Introduction

Water vapor (WV) plays an important role in the radiative balance on Earth since it is the principal source of infrared opacity. It's contribution to natural terrestrial greenhouse effect is of about 60 to 75%.

Simulations based on radiative-convective models and observations have demonstrated that an increase in greenhouse gases like $CO_2$, inducing a warming of the global surface temperature, could lead to a moistening of the troposphere (Dessler, 2013; Dessler et al., 2008; Dessler and Wong, 2009; Minschwaner and Dessler, 2004; Soden et al., 2005). This feedback effect, which has remained controversial, could double the warming induced only by $CO_2$ itself (Banerjee et al., 2019; Dessler et al., 2013). (Dessler et al., 2013) states that stratospheric water vapor (SWV) provides about +0.3 W/(m² K) climate feedback to global warming. Therefore, SWV has a great potential onto the global climate radiative equilibrium.



(Dvortsov and Solomon, 2001) have studied the response of stratospheric temperatures and ozone to past and future increases in the stratospheric humidity. From this study, it has been shown from two-dimensional radiative-chemical-dynamical model that the humidification of the mid-latitude stratosphere leads to a stratospheric cooling and the loss of ozone at these latitudes. More importantly, the impact of such cooling represents about 30 to 50% of the observed cooling.

(Solomon et al., 2010) linked the observed decrease of SWV after 2000 to a slowing of the global temperature increase between 2000 and 2009. It appears then that SWV accounts for a significant contribution to the radiative equilibrium of the stratosphere, and therefore to the global radiative equilibrium. Observational studies have shown that an increase of stratospheric water vapor could lead to a warming of the mean surface temperature (Wang et al., 2017). The equilibrium response of such perturbation seems larger over the extratropics than in the tropics. The stratospheric humidity is driven by

many complex processes and interactions for which important lack of information remains, especially about the impact of overshooting convection or the impact of increase of greenhouse gases' concentration onto the stratospheric water vapor (SWV) budget. A strong correlation between tropical convection and stratospheric humidity has been observed from observational studies. Being able to understand and to predict the evolution of the stratospheric humidity is essential to determine the future climate. Additionally, SWV plays a key role in the chemical balance of the atmosphere, especially on

the ozone layer. Indeed, ozone depletion occurring each year at polar latitudes is due to the formation of polar stratospheric clouds (PSCs), which are composed of water vapor and nitric acid. Those clouds offer a surface for heterogeneous interactions to occur. At the tropics, water vapor molecules are injected by deep convection, followed by a slow ascent, in the TTL. Once in the stratosphere, molecules are transported to the poles through the Brewer Dobson circulation. Some studies suggest that the greenhouse gases' increase could speed up the Brewer Dobson circulation (Butchart, 2014; Garcia

and Randel, 2008; Palmeiro et al., 2014) and increase the number of water vapor penetrations at the tropics. Over the last decades, numerous observational studies have been conducted in order to determine the stratospheric water vapor trend through satellite, balloon-borne observations and models. However, results from such studies lead to divergent conclusions. This is due to observational biases which are related to the technical difficulty of measuring stratospheric water vapor concentration. Some studies suggest an increase of stratospheric water vapor over the last 3 decades (Ding-Zhu et al., 2015;

Hurst et al., 2011; Rosenlof et al., 2001; Scherer et al., 2008). (Hurst et al., 2011) shows variable trends over 4 different periods between 1980 and 2010. Satellite observations from HALOE (Russell et al., 1993) (Halogen Occultation Experiment) covering the period from 1985 to 2005 show that the trend from (Oltmans et al., 2000) and (Rosenlof et al., 2001) were overestimated. Additionally, the stratospheric water vapor increase deduced from the NOAA observations are in disagreement with the cold point tropopause cooling calculated from ECMWF (European Centre for Medium-Range

Weather Forecast) field and radiosounding data. (Randel et al., 2006) suggests that the sudden stratospheric water vapor decrease observed after 2001 could be related to the simultaneous cold point tropopause cooling. (Kunz et al., 2013) demonstrates that extracting a trend around the tropopause (few kilometers above and below) is more challenging than in the higher stratosphere (4 to 10 km above the tropopause) due to the influence of global-scale changes in cold point tropopause temperature . This study used the same dataset as (Hurst et al., 2011) but does not obtain the same trend estimations. The


trend differences observed between the two studies come from the determination methods, leading results which are not comparable. (Kunz et al., 2013) also shows that the variations in stratospheric humidityobserved since 2001 is more intense in extra tropical regions than in the tropics. This could be due to changes in the lower branch of the Brewer Dobson circulation.

The stratosphere being a dry layer, measuring SWV remains a technical challenge. To combine observations from different

platforms and instruments requires to correct for the biases using chemistry climate models. This procedure may induce additional errors on the extracted trends. Additionally, biases and instrumental uncertainties reported in the literature scale from 10 to 100% at the altitude of the driest layers. Consequently, differences as high as 2 ppmv, representing50% of the typical stratospheric mixing ratio, are reported . Then, the stratospheric water vapor trend remains to date, undetermined.

In this frame, a new lightweight diode laser hygrometer, named Pico-Light $H_2O$ has been developed. This sensor has been

tested in-flight during two flights under rubber balloons from the CNES Aire-sur-l'Adour facility in France. One of this flight has occurred within the HEMERA H2020 Work package 11. For one of these flights a filament of polar air has been probed at mid-latitude. The signature of polar and mid-latitude air are observed on the water vapor and temperature vertical profiles obtained during the flight. The in-situ water vapor and temperature measurements of Pico-Light $H_2O$ have been c ompared to the Aura MLS v4 and v5 measurements for levels above 316 hPa, therefore allowing to test the improvements

made in the v5 version.

This instrument will be compared in-flight with in-situ frost-point and Lyman-α hygrometers during a balloon-borne hygrometer inter-comparison campaign to be held at the CNES balloon facility in France during the spring 2021. Additionally, it will be compared to diode laser spectrometers, frost-point and Lyman-α hygrometers at the AIDA simulation chamber during the AquaVIT-4 campaign to be held beginning of 2021. Then, the aim of this paper is to describe Pico-Light

$H_2O$ capabilities as a future potential sounding instrument for regular scientific soundings. As a test of the v5 version of the MLS data products, the present study shows the improvements brought from the v4 version from a comparison of MLS data with Pico-Light $H_2O$ measurements. The newly developed hygrometer is described in section 2 and the flight conditions in section 3. The Aura MLS data and comparison with Pico-Light temperature and water vapor measurements are discussed in sections 4, 5 and 6. The section 7 describes the observations related to the polar filament intrusion above Aire-sur-l'Adour.

**2. Pico-Light $H_2O$**

Since 2017, the lightweight hygrometer (2.7 kg) Pico-Light $H_2O$, has been developed to be launched under small rubber balloon. Figure 1 shows a detailed picture of the Pico-Light hygrometer (b) and one picture of the hygrometer during its launch under a 1200 g Totex rubber balloon from the CNES Aire-sur-l'Adour facility (France), on February 19, 2019. One of the flights has occurred within the HEMERA Work Package 11 (WP11). HEMERA H2020 (see : https://www.hemera-

h2020.eu/) is a Research Infrastructure funded by the Horizon 2020 framework Programme of the European Union. This infrastructure aims to make existing balloon facilities available to all scientific teams in the European Union, Canada and





other associated countries. The WP11 focuses on lightweight atmospheric sensors and is intended to push forward the development of small atmospheric sensors, by granting a free access to balloon and ground testing facilities.

Pico-Light $H_2O$ is the lightweight successor of Pico-SDLA $H_2O$ which has been thoroughly tested and validated over other
hygrometers in-flight and in atmospheric simulation chamber (Behera Abhinna K. et al., 2018; Berthet et al., 2013; Durry et al., 2008; Fahey et al., 2014; Ghysels et al., 2016; Korotcenkov, 2018). Based on the same technique, Pico-Light $H_2O$ is a mid-infrared diode laser spectrometer. A comparison between main characteristics of Pico-SDLA $H_2O$ and Pico-Light $H_2O$ is shown in table 1.

**Table 1 : Comparisons of the mass, acquisition and energy budget of Pico-SDLA $H_2O$ and Pico-Light $H_2O$,**
**demonstrating the improvements brought..**

|  | Pico-SDLA $H_2O$ | Pico-Light $H_2O$ |
|---|---|---|
| **Total mass (kg)** | 10 | 2.75 |
| **Electronic mass (kg)** | 2.5 | 1.4 |
| **Optical cell mass (kg)** | 3.5 | 0.7 |
| **Data points per spectra** | 256 | 1024 |
| **Energy budget (Wh)** | 9 | 3.5 |

This technique uses direct absorption spectrometry to retrieve atmospheric mixing ratios. The novelty relies on a newly developed electronic box (weight : 1.35 kg), an enlighten mechanical structure (800g) and an improved energy management. Then, Pico-Light $H_2O$ differs from Pico-SDLA $H_2O$ on its weight, sensibly smaller (2.7 kg instead of 10 kg) and its
autonomy. Indeed, telecommand/telemetry is not necessary for its operation: all necessary parameters are programmed onboard.

### 2.1. Spectroscopy

The mixing ratio is extracted from the atmospheric spectra using a non-linear least squares fitting algorithm applied to the measured line shape. The Beer-Lambert law describes the absorption. The molecular line shape is modelled using a Voigt
profile (VP). We found that fitting the VP to the measured spectra yielded residuals consistent with the instrument noise. No evidence of systematic residuals caused by higher-order line shape effects were observed for stratospheric pressures (our region of interest). Figure 3 shows an example of three atmospheric spectra of the $H_2O$ $2_{02} \leftarrow 1_{01}$ line recorded during the flight on February 19, 2019 from Aire-sur-l'Adour, at different altitudes in the lower stratosphere (39.40 hPa≡21.8 km; 82.90 hPa≡17.1 km; 142.90 hPa≡13.7 km). Each atmospheric spectra is recorded over 10 milliseconds and the frequency is



centered on the probed line. For the two flights considered in the present study, the cold point tropopause (CPT) is located around 205 hPa (11.5 km). From the ground to the balloon float altitude, the water vapor concentration varies by several orders of magnitude, from about 4 ppmv in the stratosphere to several thousands of ppmv at ground level. Then, water vapor measurements are realized by recording atmospheric spectra of two water vapor transitions, the $2_{02} \leftarrow 1_{01}$ and the $4_{13} \leftarrow 4_{14}$ lines, each of them suitable for a given range of concentration. For measurements from the ground to around 260 hPa, the

$4_{13} \leftarrow 4_{14}$ $H_2^{16}O$ line at 3802.96561 cm$^{-1}$is used. Above 260 hPa, the $2_{02} \leftarrow 1_{01}$$H_2^{16}O$ line at 3801.41863 cm$^{-1}$ is used (Durry and Megie, 2000). Both sets of line parameters are obtained from the HITRAN 2016 database (Gordon et al., 2017). In HITRAN, the line intensities for these two lines are given with a relative uncertainty of less than 1%. The water vapor transition and spectra detuning are determined prior to the launch, thus allowing for automatic selection during in-flight measurements. During the spectra processing, the standard deviation of the fitting residuals is calculated. This acts as a quality criteria of the

spectra fitting. Only the retrievals associated with  a standard deviation within the measurement noise are conserved. In 2021 - 2022, laboratory calibration against national humidity standards are scheduled in static and dynamic regimes and variable temperatures, in collaboration with LNE-Cnam and LNE-Cetiat (French national metrological institutes).

### 2.2.  Spectra acquisition and laboratory testing

The electronics is split into three bodies: brain, acquisition and control/command. The brain drives the acquisition and

control/command.

The acquisition drives the photodiode and diode laser Peltier's temperatures, as well as the generation of the current ramp, the gain and the analog signal chain. Three signals are recorded : the direct atmospheric signal, containing the molecular absorption signature, the ramp signal and the differential signal. the three signals are digitized using an ADC on parallel bus interfaced with a microcontroller.

To obtain the differential signal, the ramp signal is adjusted using a gain in order to obtain a sloping background similar to the atmospheric signal. This is the reference signal. Then, the reference signal is substracted from the atmospheric signal. The differential signal is used to check for the line peak position within the scan during the flight. A control loop is programmed to correct for the peak position in case a drift too large is detected.

The laser frequency is scanned across one of the resonance frequencies of an absorbing molecules by tuning the diode

frequency by ramping its current keeping the Peltier element at a constant temperature. The laser light is then partially absorbed by ambient molecules over the optical path length in air. The signal is detected onto a cooled photodiode and consists of a sloping background surimposed to the absorption of the target molecule. The generation of the ramp is performed using a 16 bits High Speed Digital to Analog Converter controlled by an SPI bus. One spectrum of 1024 datapoints is obtained within 20 ms, which allows for high-resolution measurements. Five spectra are recorded within the

first 200 ms. Over the following 800 ms, the measurements from the temperature and pressure sensors are recorded and the treatment of the information is realized. Then, the vertical resolution during the descent under parachute is of about 2 to 5 meters since we consider that one measurements is realized over one second. This measurement is composed of five





digitized ambient spectra and the measurements of all necessary physical and technical parameters which are useful for post-processing, analysis and technical checks.

Major issues which could be encountered while the instrument is flying are : 1- a frequency drift of the laser emission due to the cold ambient temperature, 2- from point 1, the loose of the absorption line which could have drifted out of the scanning range (in this case, no measurements could be performed) and 3- vibrations of the instrument's mechanical structure leading to severe deformations of the spectra. Each of the three cited issues have been addressed with the help of laboratory testing. At ambient temperature, the laser diode frequency has been calibrated using a mid-infrared wavelength meter which has

allow to determine the proper center Peltier temperature and current. Then, tests in cold chamber have been performed to reproduce cold temperature environments, similar to atmospheric conditions. From these tests, were determined enclosure's temperature corrections to limit the laser frequency drift due to cold ambient temperature. The correction have then, been programmed in the software onboard. Additionally, the diode laser and photodiode enclosures have been designed based on thermal studies to limit the cooling of the devices. The laser diode and photodiode temperatures are stabilized within better

than 0.01°C. The laser diode temperature is set between -5°C and +30°C, allowing to select the emission center wavelength. Indeed, the line frequency is obtained for a diode laser temperature of 15°C for the $4_{13} \leftarrow 4_{14}$ $H_2^{16}O$ line at 3802.96561 cm$^-$1 and 19°C for the $2_{02} \leftarrow 1_{01} H_2^{16}O$ line at 3801.41863 cm$^-$. A 1°C variation of this temperature allows to scan over 0.3-0.4 cm$^{-1}$/°C.

The stabilization of the laser temperature is of importance since it allows the stabilization of the center wavelength frequency

and therefore, permits to average, if needed, multiple spectra to enhance the measurement precision and detection limit.

Adjustments and improvements have been realized since Pico-SDLA H2O on the electronic architecture, and energy consumption. First, the size of the card itself has been drastically reduced. Flights under rubber balloons last 2 hours in total whereas flights under regular open stratospheric balloons last around 4 to 6 hours. Then, the weight of battery packs can be reduced by a large amount. In the case of Pico-Light $H_2O$, the total mass of batteries is of 400 grams whereas 2.5 kg of

batteries was necessary for a flight of Pico-SDLA $H_2O$ under open stratospheric balloon. Since the electronic card has been improved, and that one flight under rubber balloon is shorter, the energy consumption during one flight of Pico-Light $H_2O$ is of about 3.5 Wh, whereas it was of 9 Wh in the case of Pico-SDLA $H_2O$.

During the flight, no control of the hygrometer is necessary. The embedded software allows the hygrometer to work fully standalone with no human control. It manages the whole flight until landing. During ascent and descent, the software adjusts

the amplitude of the current modulation and also selects the proper water vapor line for the sounding based on pressure measurements. The knowledge of the spectroscopic characteristic of the two lines selected for the sounding, allows to determine which of the two lines is the most appropriate, based on the peak absorbance and the pressure level being sounded. This choice is fixed in the onboard software. Additionally, the pressure measurements allows to detect whether the balloon is ascending or descending based on the variation of its value over 15 measurements. In case an anomaly on one value of the

technical parameter is observed, the onboard software turns off the servo loop and set this parameter back to its initial value. This acts as a protection against the failure of a controlling loop.



### 2.3. Optical cell

Water vapor is measured over a 1-m path length in ambient air. The ambient temperature is measured using two Sippican VIZ meteorological temperature sondes with a precision of $\pm$ 0.1°C. The hygrometer is equipped with a PTU (pressure, temperature, humidity) sonde from InterMet (iMet-4), to compare humidity measurements in the lower troposphere. Pico-Light $H_2O$ is equipped with 1 GNSS (Global Navigation Satellite System) localization systems. A second GNSS system is found with the Imet-4 onboard Pico-Light. The environment pressure is measured using a Honeywell PPT2 absolute pressure transducer (precision : ~0.03%).

The optical elements are heated using heat resistors to limit the formation of ice or dew.

The instrument structure is made of carbon fiber tubes which presents a high stability with temperature and resistance to dynamical constraints encountered during a flight. Carbon fibers are interesting for their high strength to weight ratio, stiffness and resistance to corrosion. The mechanical design has been slightly changed since Pico-SDLA $H_2O$. Figure 4 shows Pico-SDLA $H_2O$ during the TRO-Pico campaign in 2012 (left) and Pico-Light $H_2O$ in 2019 (right). In the case of Pico-SDLA $H_2O$, the mechanical structure was made of 6 carbon tubes, tighten using 3 large pieces made of PVC. The structure of Pico-light $H_2O$ is made of 3 carbon tubes, tighten using 2 small pieces in aluminum to guarantee for the mechanical stability. The modifications in the mechanical design from Pico-SDLA $H_2O$ and Pico-Light $H_2O$ have allowed to enlighten the optical cell from 3.5 kg to 700 grams.

### 2.4. Uncertainties

The noise level over 200 ms is of about $5.10^{-4}$ in absorption units. In the stratosphere, the signal-to-noise ratio of the spectra is of about 2000. The precision is calculated from the standard deviation at one given pressure level. The response time of the pressure sensor is of about 1 second. Over one second, five spectra are recorded onboard. The standard deviation in the stratosphere, and therefore, the precision, is of about 277 ppbv for a 200 ms integration time which corresponds to a precision of 130 ppbv for a 1 second integration time. By comparison, the precision of the NOAA frost-point hygrometer is of 180 ppbv for a 1-second integration time. The optical cell design being simple, the variability of the spectra baseline is minimized. The error on the retrievals, induced by the baseline variability, has been estimated to count for less than 0.5%. Taking into account all sources of error, including spectroscopy, the spectra quality, the baseline determination error and the error on physical parameters, and summing them in quadrature, lead to a total error expected of at worst 3.5% within the UT-LS.

Infrared diode laser spectrometers like Pico-Light $H_2O$, offer several advantages compared to other in-situ closed-cell techniques. The fast acquisition and high response speed (typically few milliseconds) allow for the tracking of fast humidity changes since it is free of the need for equilibrium with the sounded medium. The open-path configuration allows to avoid contamination of the measurements by the desorption of water vapor from the walls of a closed box. Then, since it is based on non-conversion method, biases related to any contribution of the concurrent conversion of other specific species are



erased. Optical hygrometers are also highly sensitive even at low mixing ratios. Additionally, Pico-Light $H_2O$ is able to

provide a mixing ratio profile from the ground to the float altitude whereas other methods like, for example fluorescent hygrometers, are limited to pressures below 300 hPa. The opto-electronic components being long life-time, the instrument is therefore budget friendly and any damaging costs would be related mainly to the mechanical structure and then, relatively small.

### 3.    Launch conditions

Two technological flights of Pico-Light $H_2O$ took place on February 19, 2019 and October 16, 2019 from the CNES balloon facility at Aire-sur-l'Adour (France). The flight on October 16, 2019 occurred within the HEMERA WP11. The hygrometer was flown under 1200g Totex rubber balloon. The flight train included the balloon, a cutter, a parachute ($3m^2$) and the Pico-Light hygrometer.

The balloons have reached the ceiling at 27.39 km and 28.96 km on February 19 and October 16 respectively. The ascents

have last 1h30 at 5m/s. Once at the maximum altitude, the balloon burst and the payload starts a descent under parachutes during 45 minutes. Figure 5 shows a picture of the hygrometer in the recovery area after landing. No damages have been observed after landing.

### 4.    Aura-MLS temperature and water vapor retrievals

Observations of water vapor profiles from the Microwave Limb Sounder (MLS) instrument, launched on NASA's Aura

satellite in 2004, have been used in order to test improvements brought to the v5 version, in comparison with the v4.  Here we use both the MLS version dataset (Livesey et al., 2018) and the newly-released version 5 dataset (Livesey et al., 2020). At the time of writing, the MLS v5 dataset is under evaluation, and reprocessing of the entire 15-year MLS data record is yet to complete.  The main differences for water vapor product are a reduction of an estimated 20% dry bias below the tropopause, and partial amelioration of a slow positive drift seen in comparisons between MLS and other observations of

water vapor in the years since 2010.  The extent to which this reduces the drifts reported by (Hurst et al., 2016) remains to be investigated. The table 2 lists the resolution, the precision for the MLS pressure levels between 22 and 316 hPa and the corresponding precision of Pico-Light $H_2O$.

**Table 2 : Resolution and precision of water vapor MLS data and precision of Pico-Light $H_2O$ on the MLS pressure level.**

| Pressure (hPa) | Resolution (VxH in km) | Precision (%) | Pico-Light precision (%) |
|---|---|---|---|
| 22 | 3.2x265 | 5 | |
| 46 | 3.2x230 | 5 | 10 |





| 68 | 3.1x190 | 5 | 5 |
| 83 | 3.1x190 | 7 | 2.6 |
| 100 | 3.0x198 | 15 | 2.6 |
| 121 | 2.6x193 | 20 | 5 |
| 147 | 2.3x188 | 20 | 5 |
| 178 | 1.7x183 | 25 | 5 |
| 215 | 1.6x178 | 40 | 2 |
| 261 | 1.4x173 | 35 | 2 |
| 316 | 1.3x168 | 65 | 2 |

## 5.  Method for inter-comparison with Aura-MLS retrievals and selection criteria

Since the acquisition time of spectra is quick (about 10 ms), Pico-Light response time is linked with the time needed to detect a change in concentration based on the response time of the pressure and temperature sensor which brings input physical parameters to the atmospheric spectra processing. Here, the response time is therefore of about 1 second. Considering a balloon mean descending rate of 7 m/s over the full descent, the mean vertical resolution of Pico-Light

retrievals is of about 7 to 10 m. At contrast, the MLS v4 and v5 vertical resolution is of few km for water vapor and temperature. Then, the Pico-Light profiles must be resampled to match the MLS v4 and v5 vertical grid. We applied a linear interpolation of the Pico-Light profile in the pressure log space and appropriate averaging kernels as in (Yan et al., 2016). Once the Pico-Light profile has been interpolated onto MLS pressure levels, we applied the averaging kernels following the operation:

$$\widehat{X_S} = X_{ap} + \left[\overline{X_S} - X_{ap}\right]A$$

Where $\widehat{X_S}$ is the resulting sonde profile, $X_{ap}$ the apriori profile from MLS, $\overline{X_S}$ the sonde profile sampled at MLS resolution and A the averaging kernel matrix. The resulting profile in then appropriate for comparison with MLS profiles. Here, the Pico-Light data have been convolved with v4 averaging kernels for MLS v4 and v5. The averaging kernels used here are publicly available and have been derived for high latitudes.

The MLS data have been obtained in the geographical grid box of ± 2° latitude and ± 3° longitude and within 24 hours of the

flight time. For both February 19 and October 16, 2019, the MLS profiles are within 300 km (2.5°) of the launch site. Flights of Pico-Light occurred 8 hours after and 17 hours prior the MLS path respectively. For the flight on February 19, 2019, maps of water vapor retrievals from MODIS level 2 atmospheric profiles in a polygon covering 18° x 20°, including the MLS profiles and balloon's position, have been used to select the profiles probing the same air mass as Pico-Light. On February





19, 2019, 4 profiles of MLS have been found. On October 17, 2019, 2 profiles of MLS have been found nearby the balloon's

position within 24 hours. In (Vömel et al., 2007), the overpass criteria was set to 300 km and 6 hours. Indeed, larger overpass criteria lead to  an increase possibility of larger atmospheric exchanges between observations. However, the authors have tested relaxed criteria up to 900 km and 12hours without having observed changes in the comparison results. Constraining overpass criteria in the present study lead to a decrease of sample size (only one or two MLS profiles were found) for the comparison but without improving the comparison. Then, we decided to keep relaxed criteria set as ± 2° in latitude, ± 3° in

longitude and 24 hours time interval. The table 3 resumes the proximity criteria for each MLS profiles used in this study.

**Table 3 Proximity criteria of the MLS profiles used for the comparison of each flight of Pico-Light H$_2$O.**

| Criteria | February 19, 2019 | | | | October 17, 2019 | |
|---|---|---|---|---|---|---|
| Profile number | #1 | #2 | #3 | #4 | #1 | #2 |
| Distance to Pico-Light (km) | 87 | 116 | 254 | 284 | 94 | 112 |
| Time difference with Pico-Light (hrs) | -7.94 | -7.94 | +3.18 | +3.18 | +16.51 | +16.53 |
| Quality criteria | 1.79 | 1.79 | 1.84 | 1.87 | 1.67 | 1.64 |

Relative differences are calculated respectively with Pico-Light measurements from the equation :

$$\delta(P) = \frac{X_{MLS}(P) - \widehat{X_S(P)}}{\widehat{X_S}(P)}$$

Where X$_{MLS}$ are measurements from MLS at one given pressure level P and $\widehat{X_S}(P)$ is Pico-Light resulting measurement at the same pressure level.

## 6.    Results and discussion

### 6.1.    Temperature

Figure 6 and Figure 7 show the comparison between temperature profiles from Pico-Light H$_2$O and Aura-MLS v4 and ERA

5 (ERA 5, 2017). For scientific analysis, only the pressure range from 261 to 0.002 hPa is recommended for MLS. Figure 6 also displays the temperature profile from a RS92 radiosonde, launched on February 19, 2019 at 10:00 UTC, 1h 15 after Pico-Light.

The biases are calculated as :

$$\delta T = T_{pico} - T_{other}$$



Where $T_{pico}$ is the temperature measured by Pico-Light and $T_{other}$ is the temperature measured by MLS or ERA 5. The Table 4

gives the mean bias between MLS v4, ERA 5 and Pico-Light between 20 and 261 hPa. In the stratosphere, the mean bias with MLS is of about 2 K. The temperature measured by MLS is colder than the ones measured by two Sippican thermistors onboard Pico-Light for levels above 100 hPa on February 19, 2019 and above 70 hPa on October 16, 2019.

**Table 4: Temperature bias between Pico-Light, MLS v4 and ERA 5 measurements in the mid-latitude lower stratosphere.**

| Date | Temp. mean bias (K) | | | |
|---|---|---|---|---|
| | MLS v4 | | ERA 5 | |
| February 19, 2019 | 20-261 hPa | 2.05 | 70-250 hPa | -0.57 |
| | | | 30-70 hPa | -2.50 |
| October 16, 2019 | 20-261 hPa | 2.44 | 70-250 hPa | -0.85 |
| | | | 30-70 hPa | -0.09 |


ERA 5 is an ECMWF reanalysis of the global climate combining model data with observations. At ECMWF, every 12 hours, a previous forecast is combined with available dataset to generate the new estimate of the atmosphere. The mean differences between Pico-Light and ERA 5 hourly reanalysis temperature are rather small between 70 and 250 hPa (lower stratosphere). They scale from -0.6 to -0.85 K on February 19 and October 16, 2019 respectively. Higher in the stratosphere, the

differences increase on February 19, 2019 where the mean bias is of about -2.5 K but not on October 16, 2019 where it is of about 90 mK for levels above 70 hPa. The temperature sondes onboard Pico-Light are Sippican Mark II sondes. These sondes have been compared to COSMIC (Constellation Observing System for Meteorology, Ionosphere, and Climate) temperature data along with other sondes in various conditions (Sun et al., 2013). The COSMIC post processed temperature data are obtained from calculations by a fixed processing algorithm based on GPS satellite observations. In (Sun et al.,

2013), Sippican Mark II sonde exhibit a cold bias (about -0.5K) between 250 and 70 hPa compared to COSMIC. This cold bias gradually decrease toward the stratosphere to become warm bias for levels above 50hPa. In (Sun et al., 2013), the Sippican Mark II biases scale from -0.25 to 0.19K depending on the solar elevation angle and are within the sondes having the smaller biases compared to COSMIC, considered as a reference dataset in the cited study. (Marlton et al., 2020) have reported differences between temperature LIDAR from the NDACC (Network for Detection of Atmospheric Composition

Change) and ERA 5 within ± 5K between 100 and 0.7 hPa.

The observed differences between Pico-Light and ERA 5 temperatures are similar to other published values, particularly (Sun et al., 2013), on October 16, 2019. On February 19, 2019, the observed bias exhibit a different behavior regarding to altitude which could be related to the presence of a polar filament over the flight location which may not have been seen by the ERA 5 reanalysis. The differences with Aura MLS temperature products scale between 0 and 5K for levels above 261





hPa. Similar biases have been observed between Aura MLS v4 and radiosondes (RS80 and RS92) in (Yan et al., 2016). In this study, the biases scale between 2 and 6 K between 10 and 316 hPa.

## 6.2. Water vapor

Pico-Light $H_2O$ measured water vapor during the ascent and the descent.  However, ascent measurements are oftenly affected by water vapor out-gassing from the balloon envelope. Consequently, only measurements during the descent are
considered for analysis.

Figure 8 and Figure 9 show high resolution profiles of water vapor mixing ratio on February 19 and October 16, 2019, based on Pico-Light $H_2O$ measurements near Aire-sur-l'Adour (France) during two technological flights (black). These profiles are compared with MLS v4 and v5 retrievals collected near Aire-sur-l'Adour on February 19 and October 17, 2019. The profiles were interpolated on the MLS pressure grid and averaging kernels at 70°N were applied to obtain consistent vertical
resolution between the two datasets.  The resulting Pico-Light profiles are shown with open black squares. The profiles used in this analysis have been screened using the quality control criteria suggested in the v4 quality document. On February 19 and October 17, the quality criteria is around 1.80 and 1.67 respectively. The threshold criteria for scientific use is defined as 0.7. The MLS quality criteria indicate the quality of radiance fit. A value close to zero indicate a poor radiance fit and then less trustworthy data. The value of the quality criteria is used a s a threshold for rejecting data for scientific use.


For scientific analysis, only pressure levels between 316 and 0.002 hPa are valuable, as stated in the MLS v4 data quality document. Above the CPT, the agreement between in-situ and MLS profiles is the best.

The mean relative difference between MLS v4 and Pico-Light profiles is within 20% for levels above 200 hPa where the MLS precision is the best (around 0.3 ppmv ~ 5-10%, pressure below 21/5 hPa) and get larger below. Then, vertical
structures are not captured by MLS due the low vertical resolution and the uncertainties quickly increase (reaching close to 1 ppmv at 260 hPa ~ 17%). Table 5 gives mean relative differences in $H_2O$ mixing ratio between MLS v4, v5 and Pico-Light measurements between 20 and 316 hPa. The mean relative difference between MLS v4 and Pico-Light is of -7.28% and -18.79% on February 19 and October 16, 2019 respectively for levels above 200 hPa. Around the tropopause, MLS v4 exhibits dry biases compared with Pico-Light. The CPT is located at around 197 and 215 hPa on October 16 and February 19
respectively. For the highest tropopause (October 16) larger discrepancies are observed between Pico-Light and MLS v4 for pressure levels between 200 and 316 hPa. The mean relative differences are of -47.64% and -63.54% on February 19 and October 16, 2019 respectively. Similar behavior has been observed when MLS has been compared to frost-point hygrometers (Livesey et al., 2018). It has to be noted that MLS vertical profile on October 16, 2019 is not available. Then, the profiles on October 17 have been used although within 24 hours of the flight of Pico-Light (22 hours afterward).
The observed differences between Pico-Light and MLS v4 are of the same amplitude as those published in the frame of comparison between MLS and other in-situ techniques. (Yan et al., 2016) have compared in-situ balloon-borne measurements of water vapor and temperature with MLS v3 and v4 retrievals above the Tibetan plateau. Reported

differences are within ± 20% at levels above 100 hPa. For levels below 100 hPa, differences can reach 80%. (Herman et al.,
2017) have compared in-situ airborne observations from the laser hygrometer JLH Mark 2, onboard the ER-2 aircraft, with

MLS over North America between 15 and 19 km. The reported differences can reach as high as 2.5 ppmv (~46%) at 14.5 km
(about 125 hPa) and are of about 1 ppmv (~ 25%) at 19 km (about 60 hPa). (Sunilkumar et al., 2016) have compared in-situ
measurements of the NOAA cryogenic frost-point hygrometer with water vapor retrievals from MLS between 10 km (~260
hPa) and 24 km (~20 hPa) over India. The differences are of about 1 ppmv (~20 %) at 20 hPa with a minimum of 0.2 ppmv
(~7%) at 80 hPa reaching a maximum of 100 ppmv (~40 %) in the upper troposphere.

At 82 hPa, the mean difference between Pico-Light and MLS v4 is of 0.247 ppmv on February 19, 2019 and of -0.251 ppmv
on October 16, 2019. Published studies in the frame of the SPARC water vapor assessment (Jensen et al., 2011) have shown
differences between the NOAA frost-point and MLS of about 0.18 ppmv over Boulder (CO, USA) and 0.11 ppmv in Lauder
(New Zeland). Differences with the FISH hygrometer scaled from 0.02 to 0.474 ppmv, those with the Harvard Lyman-α
scaled from 0.73 to 1.6 ppmv and from -0.3 to 1.4 ppmv with the JPL hygrometer. Then, the comparisons between Pico-

Light and MLS v4 are in-line with other studies.

Recently, the release of the v5 MLS data has allowed the comparison of Pico-Light measurements with MLS v5 water vapor
retrievals. As stated previously, the v5 MLS water vapor data have been corrected to reduce the dry bias below the
tropopause and to ameliorate the slow positive drift which has been observed in comparisons with other observations during
the 2010s. Then, resulting relative differences between Pico-Light and MLS v5 retrievals are shown in Figure 7 and Figure

8, together with v4 results for comparison. The v5 processing has allowed to reduce the bias nearby the tropopause, as
expected, reducing from -63.54% in average for the v4 to -50.96% for the v5 on October 16, 2019 and from -47.64% for the
v4 to -37.98% for the v5 on February 19, 2019. However, for both flights, we can note that the stratospheric retrievals are
shifted by + 8%, expected as part of the correction for the drift in MLS water vapor implemented as part of the MLS v5
algorithms.

**Table 5: Mean relative difference in water vapor mixing ratio between Pico-Light and MLS (v4 and v5), between 20 and 200 hPa
and between 200 and 316 hPa for both Pico-Light flights.**

| Date | H$_2$O mean rel. diff (%) | | |
|---|---|---|---|
| | | **MLS v4** | **MLS v5** |
| **February 19, 2019** | **20-200 hPa** | -7.28 | -15.50 |
| | **200-316 hPa** | -47.64 | -37.98 |
| **October 16,** | **20-200 hPa** | -18.79 | -26.21 |





| 2019 | 200-316 hPa | -63.54 | -50.96 |
|------|-------------|--------|--------|

The differences between Pico-Light and MLS datasets are included within Aura-MLS uncertainties for levels above 200 hPa. In this case, in the stratosphere, timescale of exchanges between air masses is larger than levels below. For levels below 200 hPa on February 19, 2019, the differences arise from large MLS uncertainty and from the particular meteorological conditions: the presence of a polar filament crossing southern France from February 18 to February 19, 2019. The vertical resolution and the measurement uncertainty of MLS does not allow to capture the subtle changes in the water vapor mixing ratio related to the filament.

On October 16, 2019, the differences between Pico-Light H2O and MLS are globally larger. This is due to the largest time difference of the overpass, 16 hours instead of 7 hours in the case of February 19, 2019. Additionally, one case note that the quality criteria of the MLS data on October 17, 2019 is not as good as in the case of February 19, 2019.

### 6.3. Comparison with radiosonde and ERA 5 hourly reanalysis in the troposphere

On February 19, 2019, a Vaisala radiosonde RS 41has been launched at 10:00 UTC, 1h30 after to the launch of Pico-Light. Additionally, a PTU sonde from InterMet is included onboard Pico-Light in order to check for technical issues during the beginning and the end of the flight. Humidity measurements provided by radiosondes are valuable only in the lower troposphere (up to 400 hPa here). Figure 10 illustrates the comparison between mixing ratio measurements from Pico-Light, the Imet sonde and the RS41 between the ground and 400 hPa.

The mean difference between the humidity measurements from the Imet sonde and Pico-Light is of - 4.95 % . The Imet sonde humidity sensor is a thin-film capacitive polymer that responds directly to relative humidity. The uncertainty from - 40°C to above 0°C is given at 5%. Pico-Light uncertainty at this altitude is of 2.2%. The observed difference is then within the Imet sonde uncertainty.

The differences with RS-41 are larger, particularly between 800 and 600 hPa, due to the fact that the RS-41 has been launched earlier than Pico-Light and had slightly different trajectory than Pico-Light. In the lower troposphere, humidity is highly variable, at time scale of an hour or less. The particular meteorological conditions has induced strong layering the dry and humid air above Aire-sur-l'Adour. The Rs-41 radiosonde has probe the humid air mass at a different location than Pico-Light H$_2$O, then humid structures are more intense in the RS-41 measurements than in those of Pico-Light H$_2$O.

The ERA 5 hourly reanalysis dataset is showing a humid layer between 800 and 600 hPa and a second humid layer between 200 and 300 hPa, Figure 12 shows a cross section of relative humidity from ERA 5 hourly reanalysis centered at 43.75° in latitude and spanning a longitude section spreading from -1 ° to +5° E. The vertical axis is the pressure. The position of Pico-Light H$_2$O is represented by a solid black line. In the right, the water vapor profile of Pico-Light H$_2$O is shown for comparison. We can notice the humid layers observed with ERA5 are also visible on Pico-Light H$_2$O tropospheric measurements, at the same pressure levels. The correspondence between ERA 5 and Pico-Light humid layer is represented





by red dashed lines. One can see a thin humid layer between 230 and 280 hPa and a thick layer between 400 and 800 hPa,
which is thinner at the location of Pico-Light H₂O (spreading from 375 and 675 hPa)..February 19, 2019 flight

In January 2019, the polar vortex split into two major parts, one over Canada and one over Russia. This pattern has led to a severe winter lasting throughout February in the United States. It then turned into a wavy shape configuration. Consequently, over February 2019, several filaments of the vortex have circulated through Europe, crossing France several times. Those polar-originated filaments are associated with cold temperature with respect to mid latitude stratospheric air masses. In case of no dehydration due to PSC2 in the polar vortex (dehydration is relatively rare in the Arctic), the polar air masses are usually wetter than the mid-latitude ones. Vertical profiles of water vapor from the descent of Pico-Light on February 19, 2019 and October 16, 2019 are shown in black and grey respectively, in the left panel in Figure 11. The Pico-Light temperature profiles on February 19 and October 16, 2019 are shown in black and grey respectively in the right panel. The water vapor vertical profile obtained on February 19, 2019 exhibits vertical structures between 80 and 130hPa (17.5 and 14.4 km respectively). At this date, the instrument was flying inside a filament of polar vortex most of the flight in the UTLS, except between 80 and 130 hPa.

The decrease in water vapor is correlated to a warm pool on the associated temperature profile in the same altitude range, as an indication of mid latitude origin. The temperature profile exhibits a succession of cold and warm pools. Where Pico-Light has sounded mid-latitude air, the ambient temperature is warmer by about 2.5 K compared to altitudes where polar air has been sounded. (Müller et al., 2003) has reported similar observation over Ny-Ålesund where water vapor could be used as a tracer of vortex filamentation at the same period (February).Vertical profiles of water vapor from the NOAA/CMDL demonstrated a dry layer where mid-latitude air has been sounded associated with a warm pool on the temperature profiles.

Potential vorticity (PV) is oftenly used as dynamical tracer of air masses, particularly for polar air masses having a high potential vorticity. The semi-lagrangian advection model MIMOSA (Hauchecorne et al., 2002) has been developed at Service d'Aéronomie in the frame of the European Union project Meridional Transport of Ozone in the Lower Stratosphere (METRO). The basic assumption is that potential vorticity and ozone mixing ratio are very well correlated on an isentropic surface and the location of ozone filaments can be visualized using potential vorticity as a quasi-passive tracer.Here, the potential vorticity has been computed on isentropic levels 380 (131hPa) and 435 K (82 hPa).The potential vorticity map on the 380 K level (3 a) corresponds to an altitude where Pico-Light was flying inside the polar filament. At this altitude, Pico-Light has sounded a wetter air mass, typical of a polar non-dehydrated air mass. The level at 435 K (Figure 13 b) corresponds to an altitude where Pico-Light was flying outside the filament, leading to dryer and warmer air mass. This is confirmed by the PV maps: the white mark on the maps shows the position of Aire-sur-l'Adour. At 380 K, we clearly see that a polar filament was over Aire-sur-l'Adour but not at 435 K.



## 7. Conclusions

A newly developed hygrometer, Pico-Light $H_2O$ has been flown twice from the Aire-sur-l'Adour (ASA) CNES balloon facility on February 19 and October 16, 2019. The hygrometer has been flown under 1200 g Totex balloon bursting at float, therefore limiting the pollution of the measurements during the descent. The two flights were successful leading to continuous measurements between the float altitude (27.3 km) and ground. For the flight of February 19, 2019, the water vapor profile shows a succession of wet and dry layers associated with cold and warm pools on the measured temperature.

From analysis, it has been shown that wet and cold air masses are linked with a polar filament sounded over CNES ASA facility. Dry and warm layers are related to mid-latitude air masses. This has been confirmed from MIMOSA potential vorticity maps calculated on 380 K and 435 K levels. Then, the Pico-Light $H_2O$ hygrometer has been demonstrated to be a promising tool for atmospheric applications.

### Aknowledgements

We are grateful to Nathaniel J. Livesey and the MLS team for the fruitful discussions about MLS water vapor data and the early release of the v5 data. This work is based on observations with Pico-Light $H_2O$ under a balloon operated by CNES, under the agreement between CNES and CNRS/INSU, within the HEMERA WP11.

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

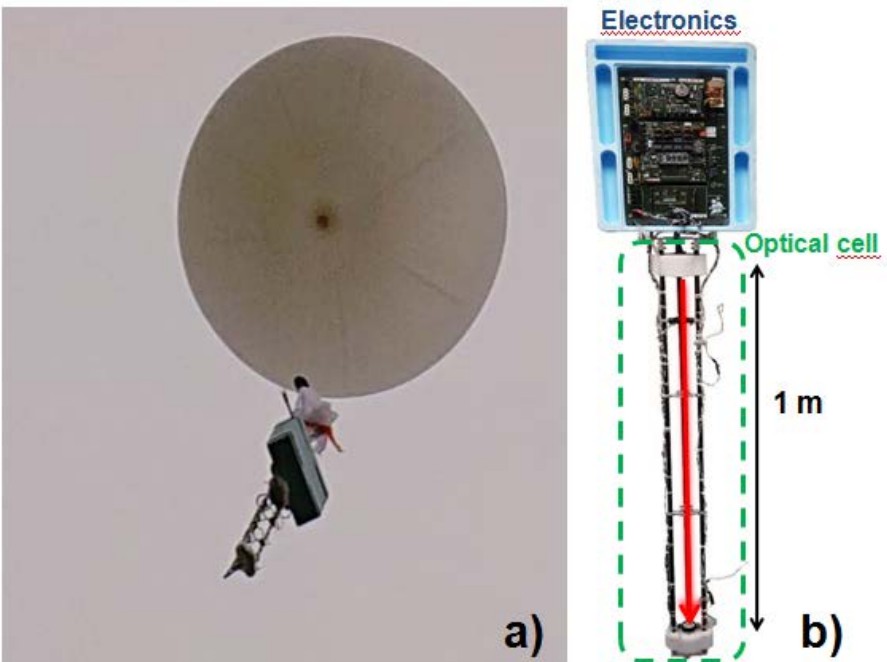

**Figure 1: Picture of Pico-Light H$_2$O during its launch under 1200g rubber balloon on February 19, 2019 from the**
**CNES Aire-sur-l'Adour facility (France) (a). On the right (b), is a detailed picture of the hygrometer**.





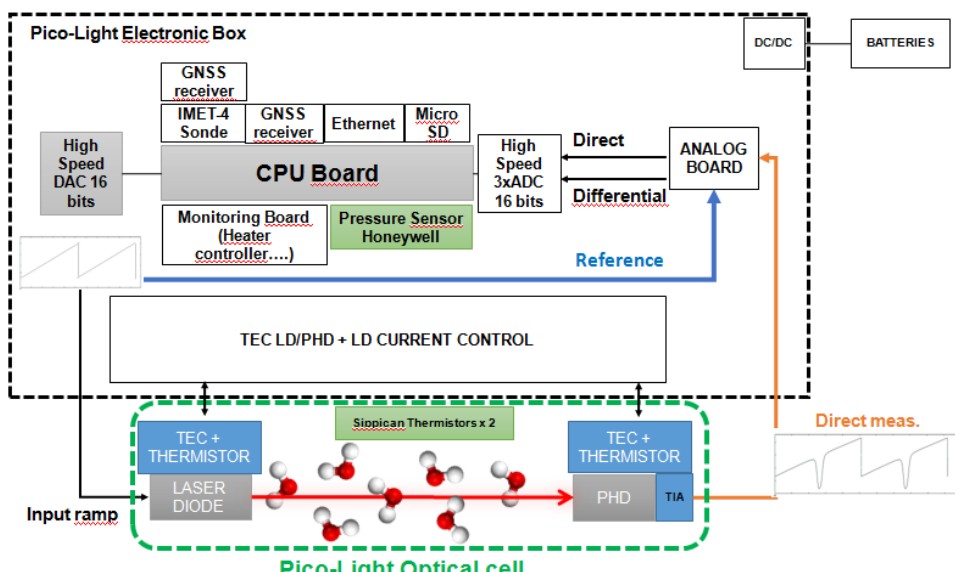

**Figure 2: Schematics of Pico-Light H$_2$O**

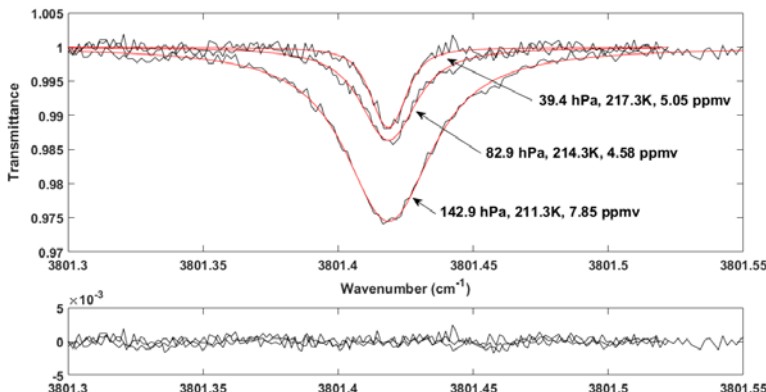


**Figure 3: Atmospheric spectra of the $2_{02} \leftarrow 1_{01}$ line of water vapor in the stratosphere (black line, top panel) together with fitting curves (red lines, top panel). The spectra have been recorded at 39.4, 82.9 and 142.9 hPa. The bottom panel shows the residuals from the fitting procedure.**



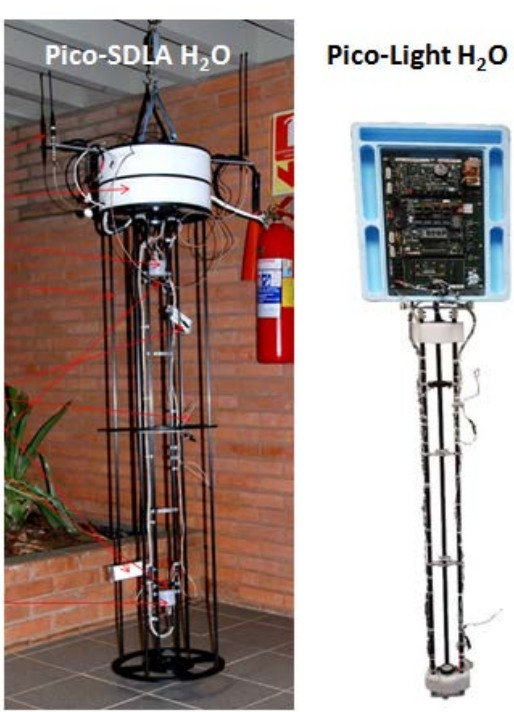

**Figure 5 : Picture of Pico-SDLA H₂O (left) during the TRO-Pico balloon campaign in 2018 (Brazil) and of Pico-Light H₂O (right) in 2019.**

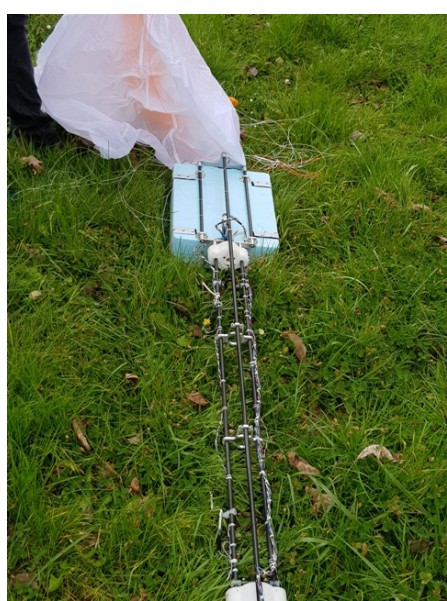

**Figure 5 : Picture of the recovery area of the Pico-Light hygrometer.**





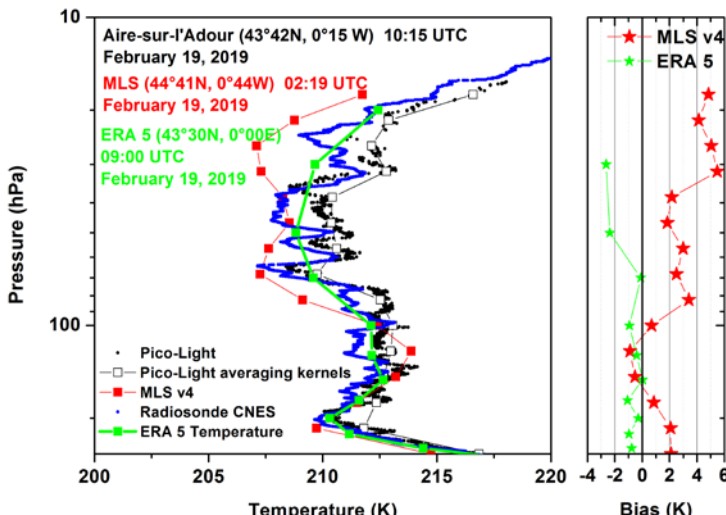

**Figure 6: Temperature profiles from Pico-Light (black) compared with Aura-MLS v4 temperature retrievals (red squares)and ERA 5 (green squares) on February 19, 2019. The open black profile is obtained by applying the smoothing function and averaging kernels at 70N onto Pico-Light profile. The right panel shows the temperature bias between Pico-Light and MLS v4 (red) and ERA 5 (green).**


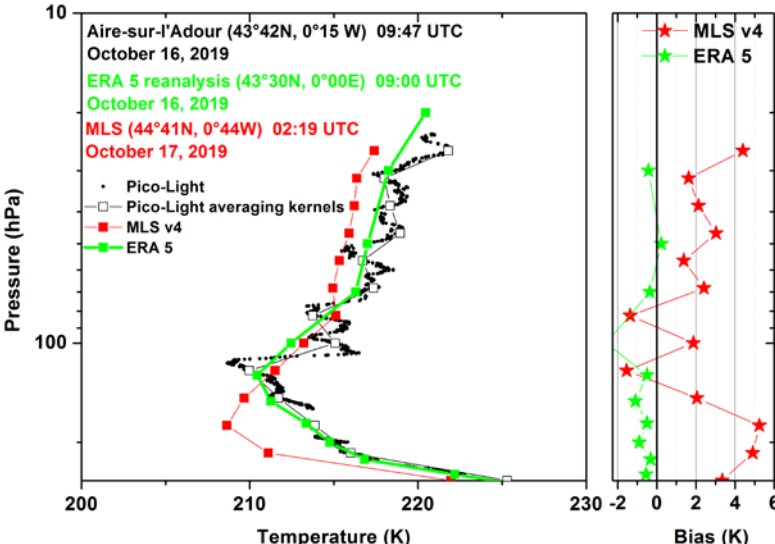

**Figure 7 :Temperature profiles from Pico-Light (black) compared with Aura-MLS v4 temperature retrievals (red squares) and ER 5 (green squares) on October 16, 2019. The open black profile is obtained by applying the smoothing function and averaging kernels at 70N onto Pico-Light profile. The right panel shows the temperature bias between Pico-Light and MLS v4 (red) and ERA 5 (green).**


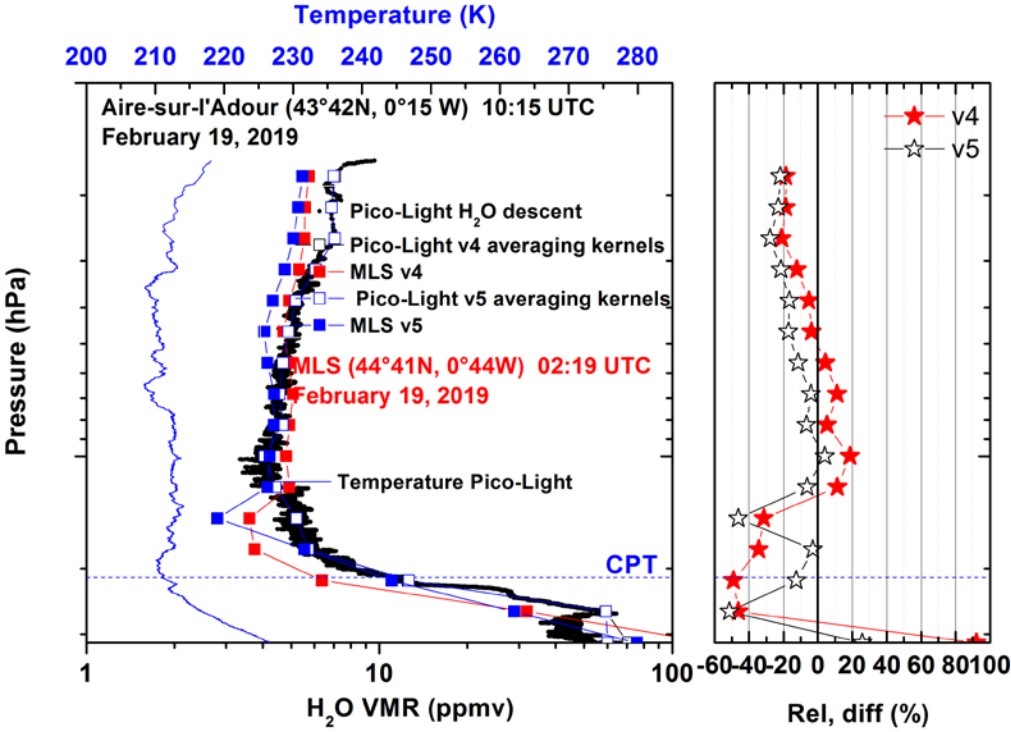

**Figure 8 : Descent high resolution vertical profile of water vapor from Pico-light H₂O (black) compared with Aura-MLS v4 (red squares) and v5 (blue squares) on February 19,2019. The open black and open blue square profilesare obtained by applying the smoothing function and averaging kernels at 70N for v4 and v5 MLS respectively.**





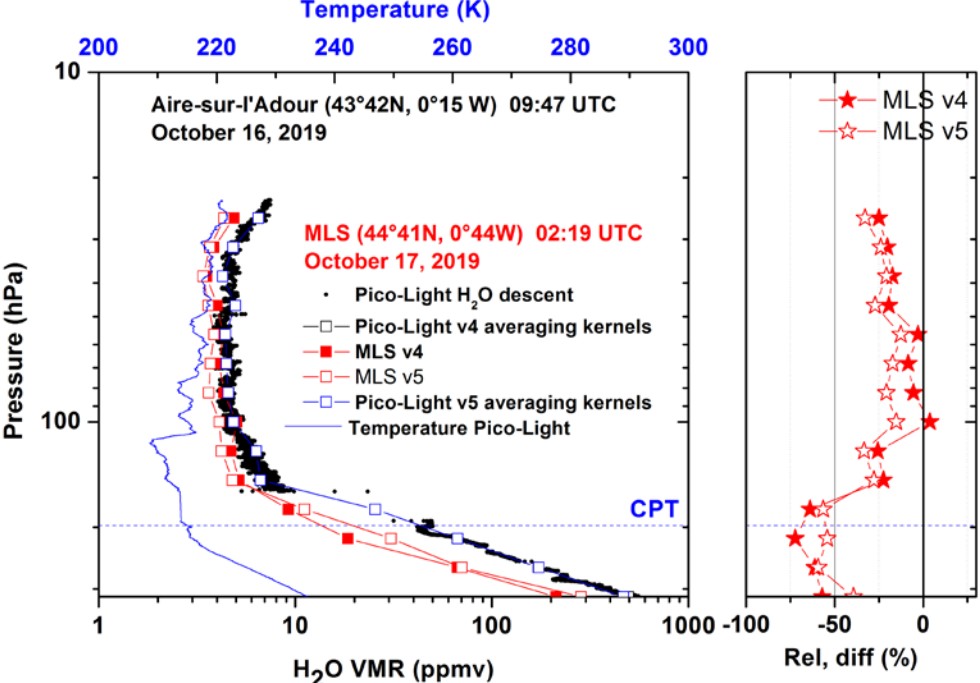

**Figure 9: Descent high resolution vertical profile of water vapor from Pico-light H₂O (black) compared with Aura-MLS v4 (red squares) and v5 (blue squares) on October 16,2019. The open black and open blue square profilesare obtained by applying the smoothing function and averaging kernels at 70N for v4 and v5 MLS respectively.**

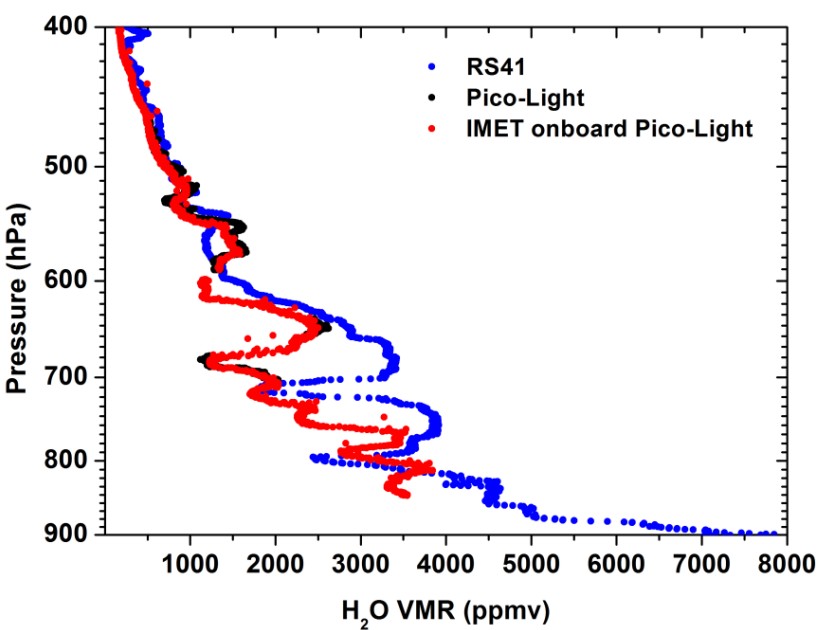



**Figure 10: Vertical profiles from the Pico-Light H$_2$O measurements (black line) compared with RS-41 radiosonde**
**operated by CNES (blue line) and IMET sonde measurements onboard Pico-Light (red line) in the lower troposphere.**

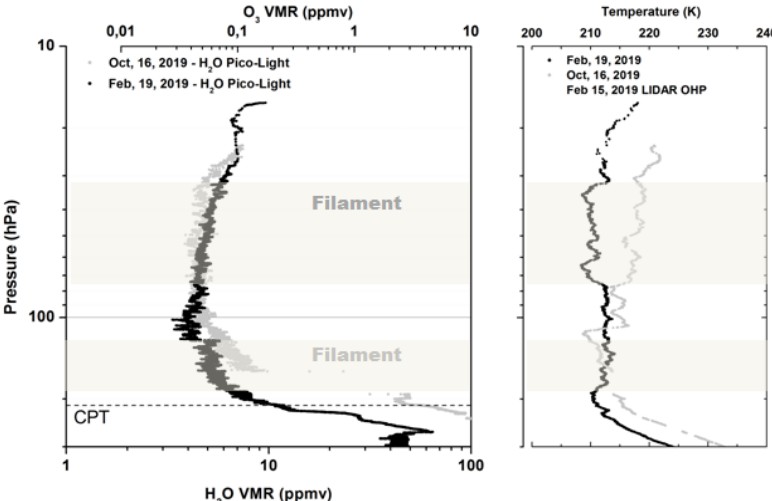

**Figure 11: Vertical profiles of water vapor from the descent of Pico-Light on February 19 and October 16, 2019 respectively the black and grey lines in the left panel. The Pico-Light temperature profiles on February 19 and October 16, 2019 are shown in black and grey respectively in the right panel.**





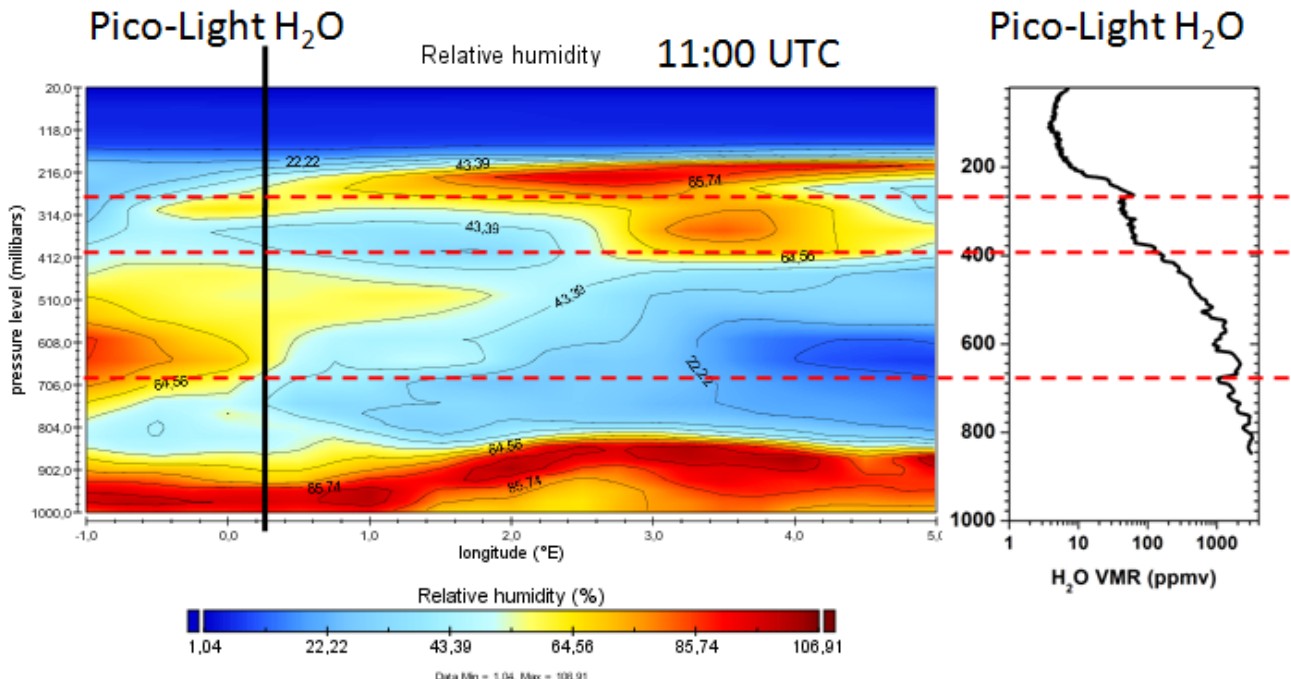

**Figure 12: Cross section of relative humidity from ERA5 hourly reanalysis at 43.75° in latitude, spreading from -1° to +5° in longitude. The Y-axis is the pressure level. On the right, the vertical profile of water vapor from Pico-Light $H_2O$ is shown for comparison.**

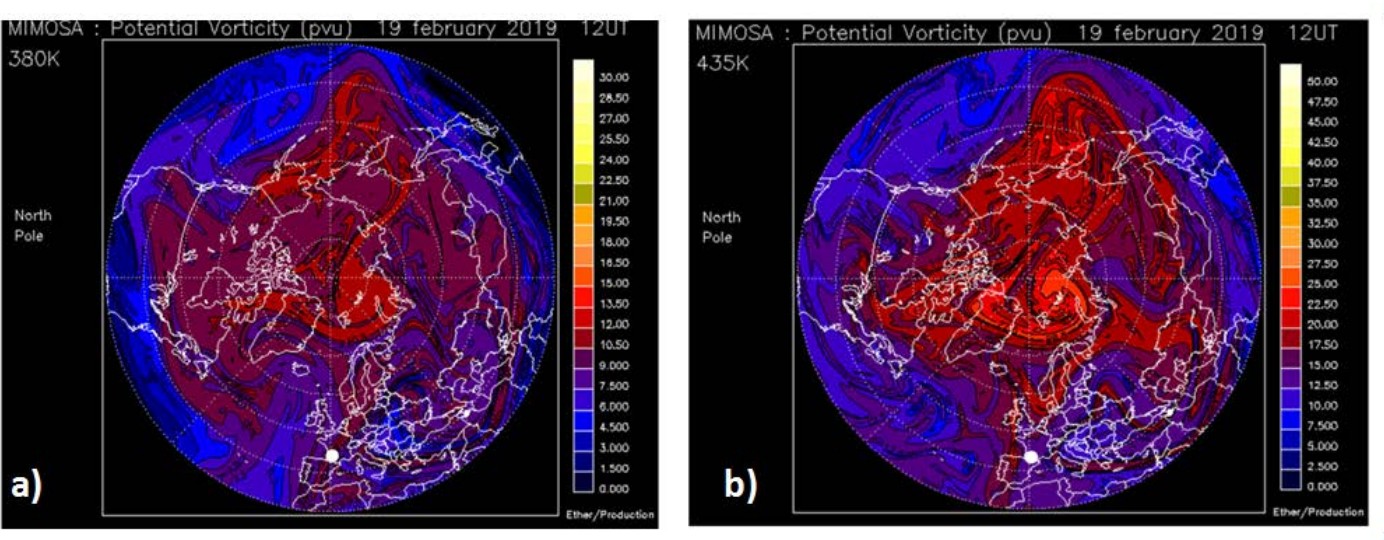

**Figure 13: Potential vorticity maps at 380 and 435 K from the MIMOSA advection model.**