# Peer review of "A lightweight balloon-borne mid-infrared hygrometer to probe the middle atmosphere: Pico-Light H2O. Comparison with Aura-MLS v4 and v5 satellite measurements"

_Atmospheric Measurement Techniques, 2020_

## Referee Comment (RC1) · Anonymous Referee #3 · 11 Nov 2020

This manuscript describes a new development based on the established Pico-SDLA laser hygrometer. The authors describe some of the significant change from the original instrument, and provide a few validating measurements based on Aura MLS satellite and radiosonde measurements.

The manuscript is an important contribution describing the next development in the line of these laser hygrometers. However, it has weaknesses that need to be addressed before publication. In particular, the description of the measurement, processing, and their associated uncertainties lacks a lot of detail. In addition, the manuscript will require significant language editing. I highlight my detailed concerns below and point out language issues, where they negatively affect the understanding.

I would recommend publication only after some major revisions.

Major comments

The validation observations, in particular the stratospheric validation does require some improvement and more instrumental discussion. Using Aura MLS is appropriate, but having just two profiles is not sufficient to provide a strong quantitative validation. Although the precision of the measurements is significantly low, atmospheric variability in the comparison of remote and in situ observation is the dominating source of uncertainty here and limits what can be deduced. There is a reasonable agreement with MLS roughly in the range of 100 hPa – 50 hPa and a significant wet bias at lower pressures. The upper tropospheric dry bias of MLS is still visible and limits what can be said about Pico-Light. Without more soundings, which apparently are scheduled, this cannot be remedied. The manuscript would gain tremendously, if the comparisons that are planned in a few months could be included here. However, even though the benefit seems obvious, I do not this it is required to include additional sounding data. What is essential is a more detailed discussion of the error sources and their vertical dependencies. I give more detailed suggestions below.

The discussion of the atmospheric filament is of marginal interest here and does not really contribute to a better understanding of the instrument. I find it a little distracting and would propose to move it from this to a different manuscript, which can focus more on the atmospheric aspects of that observation.

The discussion comparing versions 4 and 5 of MLS water vapor is also a little distracting. Pico-Light cannot contribute to evaluate which version is better. All that can be said is that the two versions agree reasonably well with the Pico-Light measurements. I would propose reducing this discussion in favor of an expanded instrumental discussion.

Detailed comments

Lines 17-18: I believe the authors refer to the difference of the two versions provided by the JPL MLS team. I do not believe that they mean that the authors applied a dry bias correction for their manuscript as this sentence implies. I would suggest just deleting it.

Line 38: A better reference for the proposed radiative feedback of stratospheric water vapor on surface temperatures is Forster and Shine (GRL 1999).

Lines 56 to 60: These two sentences require references. Which HALOE observations indicate that the Boulder trend may be an overestimate? Which study suggests that ECMWF derived tropopause temperature trends are inconsistent with stratospheric water vapor trends from the NOAA observations?

Lines 71-72: Please give references and describe where these differences have been observed to provide proper context.

Line 73: The stratospheric water vapor trend to date does not remain "undetermined". Quite the opposite. Much is known about the stratospheric water vapor trend from a multitude of instruments. However, there is still some uncertainty to the exact magnitude of the trend. It is this level of uncertainty that drives the requirements of instrumental uncertainty. The authors should spend more text here to describe where observational and analytical limits are and what these imply for the requirements on their instrument.

Lines 115 ff: The authors state here that no higher order line shape effects were observed for stratospheric pressure and state that this is their region of interest. However, there is a significant amount of discussion of lower to upper tropospheric measurements. Therefore, a discussion of higher line shape effects seems necessary.

Line 121: Do the authors mean "burst altitude"? Or is the balloon indeed floating for some time?

Lines 140ff: It is not clear to me, what the ramp or reference signal is that is subtracted from the measured spectrum. A figure could illustrate what is meant if this is done in real time. Or is there a laboratory calibration step involved to obtain the gain? Overall, I am missing a more detailed explanation of how the measurements are processed and a final mixing ratio is calculated.

Lines 149ff: One single spectrum takes 20 ms. Does that mean that there are 50 spectra measured per second potentially allowing that temporal resolution? Or is there an averaging scheme involved, that also includes the ambient pressure and temperature measurements? I would like to see, how sensitive the spectral analysis is on temperature and pressure. Would it be necessary to measure pressure and temperature and the same higher frequency to improve the temporal resolution or would it be sufficient to interpolate pressure and temperature onto the higher resolution spectral measurements?

Lines 151 ff: I assume that measurements are taken on parachute descent. Depending on the parachute size the fall velocity will vary strongly with density, most likely more than a factor 2.5. Please state what size parachute you used and show a fall velocity profile, which combined with a proper temporal resolution will give the vertical resolution.

Lines 162 ff: What temperature corrections exactly are described here? How do they contribute to the measurement uncertainty?

Line 178: I assume the authors mean that "no remote control" is necessary, which implies that the system relies on one-way telemetry only.

Could you explain whether all data needed to generate a profile are sent by telemetry or whether instrument recovery is needed to obtain high rate data?

Line 192: Is the Imet-4 just flown piggyback to get a pressure, temperature, and humidity profile from this radiosonde, or is it used to send instrument data to the ground?

Later in the manuscript you mention that a Sippican Mark-II sonde was flown as well. Was that one maybe used as radiosonde or data carrier? Can you better describe the setup of the payload?

Line 193: It would be better to give the absolute precision of the Honeywell pressure sensor. The precision is normally provided relative to the full measurement range, which was not provided. Therefore, 0.03% is not very meaningful.

Line 206: The response time of the Honeywell pressure sensor is much faster than 1 s. Maybe the authors mean sampling time? Or is there a more elaborate measurement scheme used?

Lines 206ff: Could you be more specific on the averaging? In the previous section you mentioned that the spectra a measured only for 200 ms, followed by temperature and pressure measurement and processing. If that is the case, then the uncertainty over 200 ms should be the same as over 1 s.

Lines 210ff: Please expand the discussion of the uncertainties and provide a table. You state that this estimate applies for the UTLS. What are the uncertainties above and below the UTLS? Since you show the data and the capabilities of the instrument up to 20 hPa, it is important to understand the uncertainties over the entire range.

Table 2: The precision of the measurements is not sufficient to evaluate the difference between the MLS and Pico-Light. For Pico-Light, an estimate of the possible systematic errors is important. Since there are only two profiles and no averaging can be done, the random uncertainty may be dominating, but this is not at all clear. Although the authors do not mention it, I suspect that there may be a calibration somewhere in the operation of their instrument. If not, this could be pointed out.

Line 249: See above. A parachute descent rate depends on altitude.

Lines 262: How does MODIS contribute to the analysis? This requires more explanation about the product that is used and how it is used.

Section 6.1: I am a little concerned about the temperature profiles. Are they measured by the Imet-4 temperature sensor? Are they measured on ascent or descent? Radiosondes are designed to measure during ascent and descent measurements may have reduced reliability. Or was a different sensor used, e.g. the Sippican thermistors? If so, which, how where they exposed and coated? Was there a proper radiation correction applied?

Lines 296ff: The comparison between Cosmic and the Sippican Mark-II sondes is only relevant, if these sondes were flown exactly the same as operational sondes. If they were flown piggyback as part of the larger payload with a shorter unwinder, and possibly measuring on descent, then this comparison does not apply and more discussion of exposure, ventilation, and processing is needed.

This entire paragraph does not appear to apply to the payloads that were flown and should be deleted.

Lines 322: Please explain first what the MLS quality criteria are, where they come from and what they mean, before stating the thresholds that were used.

Lines 329: What pressure is meant? 215 hPa or 21 hPa? Or something else?

Line 343 and other places in the manuscript: When referring to "levels" it would be good to specify either "pressures" or "altitude levels". For example, here, it is not clear whether you mean "for altitudes below 100 hPa" or "for pressures below 100 hPa".

Line 347: I believe Sunilkumar used the CFH, not the NOAA frostpoint hygrometer.

Lines 354f: It is hard to see the exact bias of Pico-Light in the stratosphere, but it seems to be in the range of 20 % above MLS. This is larger than the wet bias of MLS against the frostpoint instruments or the Lyman alpha instruments in that altitude range.

Line 375: delete "globally"

Lines 381 ff: This statement is not correct. The quality of radiosonde humidity measurements depends on the manufacturer and no blanket statement is possible. The Vaisala sondes are generally much better than the other manufacturers. The Imet-4 is not well validated.

I don't think there is any value including the Vaisala data here for the reason the authors provide. The collocation issue is so large that no real conclusion can be drawn. I would remove mention of that sonde.

Line 400: Something is missing in this line.

Line 430 and 433: Again: Do you mean the balloon floats before bursting?

Line 432: Replace "pollution" with "contamination"

Lines 430ff: In the conclusions, please provide an overarching discussion of the uncertainties and how they compare with the current state of technology and measurement requirements.

The reference to Jensen et al. is not a valid link. A better reference for the comparison of MLS and NOAA frostpoint is Hurst et al. (2016).

Figure 3: Are the spectra shown single spectra, or have they been averaged over a period?

Figure 5: the right panel is the same as in Figure 1. Please combine these panels?

Figure 5: Open path laser hygrometers often suffer from unwanted absorption between the laser and the exit window or fiber coupling. The same issue may be the case here. You have a Styrofoam insulation box, which most likely generates relative large water vapor concentrations inside. It is not clear where the laser light exits the laser, but I would assume that it may have to cross a window and thereby the large water vapor concentration inside the box. Is this considered and if so, how? Please spend some text describing this potential issue.

Figure 8: The legend interferes with the data. Here, as in Figure 9, I cannot make out

the black open squares.

Figure 9: Why is there a data gap just above the 200 hPa altitude? This gap also shows signs of a 5 point running mean. Please explain why this running average is needed and how it contributes to the uncertainty/vertical resolution.

Figure 10: Please remove the Vaisala RS41 data. They are more confusing than they help. Please use different symbols for the Pico-Light and Imet humidity data such that the vertical coverage of each can be seen.

Figure 11: This Figure shows two regions shaded as filament and profiles from two very different days. The upper left axis also refers to ozone, which is not shown and not mentioned in the manuscript. Highlighting the region of a possible filament has no meaning here, since the two profiles are from very different seasons. I would suggest removing Figures 11-13 from this manuscript and to expand that discussion in a separate manuscript with more care and detail.

---

## Referee Comment (RC2) · Anonymous Referee #2 · 11 Nov 2020

GENERAL COMMENTS:

This paper presents results from two flights of the Pico-Light H2O hyrgrometer, a lightweight balloon-borne mid-IR diode laser spectrometer developed at the University of Reims. The launches took place in February and October 2019 at the CNES flight facility at Aire-sur-l'Adour, Nouvelle-Aquitaine, achieving float altitudes of 27.4 and 29 km, respectively. Temperature as well as water vapor measurements for each flight are reported. (The former are presumably derived from radiosondes incorporated into the Pico-Light instrument package, though this is not stated explicitly; see Specific

Comments below). In any case, the Pico-Light measurements are compared with both Aura MLS satellite measurements as well as the ECMWF ERA5 reanalysis. In the case of water vapor, comparisons are also made with MS v5 since it incorporates significant changes from v4. All of these comparisons are presented in figures showing profiles of Pico-Light measurements overlain by the corresponding MLS and ERA5 along with profiles of sonde/MLS and sonde/ERA5 biases. Bias calculations are made after kernel-averaging the full-resolution Pico-Light profiles. (The water vapor comparisons are limited to the descent profiles to avoid potential balloon-wake contamination.)

My first concern is that the temperature comparisons add little to the understanding of the performance of the Sippican VIZ sonde, certainly not without comparison against another in situ sonde temperature.

With regard to the water vapor mixing ratio measurements, the MLS v4 water vapor comparisons suggest that the Pico-Light $H_2O$ hygrometer in these two flights is comparable to other in situ instruments. The MLS v5 comparison is of some interest, but the two flights present do not present the same story in the below-100 hPa region where v4 is understood to have an instrumental bias.

More significantly, it goes without saying that two flights – and two flights in significantly different meteorological settings - are a very slim basis upon which to make a judgement of the performance of an instrument measuring any atmospheric trace constituent, and water vapor with its strong vertical gradients particularly so. Thus I don't see great value in the profiles presented; what would be of considerably greater interest would be head-to-head intercomparison with a reference-quality in situ hygrometer.

Overall, the paper suffers from a lack of a clear purpose. While it certainly shows that the Pico-Light $H_2O$ hygrometer can make in situ measurements that are not inconsistent with other in situ water vapor instruments, it falls well short of presenting significant new findings, not withstanding the comparison to the relatively new MLS v5 water vapor.

Technically, the paper suffers a great deal from a very uneven logical and narrative flow, not to mention its need for considerable copy-editing quite beyond what can be expected from a review.

SPECIFIC COMMENTS, by section:

A. "Pico-Light H2O" (lines 84-150)

This section is poorly organized and confusing as a result. I have listed below several passages in the section that really threw me off at first and even upon re-reading, did not present a clear picture of the components comprising the instrument package, how they operate together and, most importantly, how the science data are processed:

o The section begins by restating that Pico-Light H2O is an hygrometer. However, the instrument as flown, and as reported here, provides measurements of temperature as well as water vapor mixing ratio.

o The package is assembled with two Sippican VIZ sondes are "to measure ambient temperature". That clearly suggests that these sondes are used to produce the reported "Pico-Light" temperature, but here is no discussion of how the sonde data are processed to yield the temperatures reported in the Results and Discussion section.

o A little further on it is mentioned that an (Intermet USA?) Imet-4 is included in the payload as a "PTU sonde". While it is stated that this additional sonde is for lower tropospheric humidity comparisons, the reference to it as a "PTU sonde" suggests some distinction between it and the Sippican sondes. What that distinction might be is not explained.

o Ambient pressure is measured by a Honeywell PPT2 pressure transducer; a precision of 0.03% is stated, though accuracy is not.

o There is a fairly detailed discussion of the means by which Pico-Light H2O processes the individual water vapor spectra into instantaneous (10-ms) mixing ratio values. Since no further processing is described, it must be assumed that these 100 Hz data are the

input to the kernel-averaging described three sections later.

B. "Method for intercomparison with Aura-MLS retrievals" (lines 170-189)

This begins with a discussion of the 1-Hz effective response time of the instrument resulting from the pressure and temperature transducers. While the response time of those transducers had not previously been addressed, this estimate seems reasonable. What is a bit confusing however is what bearing this has on the averaging kernel calculation that immediately follows. As mentioned in the previous comments, is it the 100 Hz profile that is input to the averaging kernel calculation? Or a 1-Hz smoothing?

C. "Results and discussion - Temperature" (lines 191-224)

This section finally provides the unequivocal statement that the Sippican VIZ sondes provide the Pico-Light temperature measurement. Although again, how the two temperature data streams are combined is not described – or at what rate the temperature is sampled.

D. "Results and discussion – Water vapor " (lines 225-279)

Here the authors present water vapor profiles in Figure 8 (?) and 7 that, as expected, show that MLS v4 water vapor mixing ratios are on the order of 20% and lower than the hygrometer at the higher stratospheric levels, come into closer agreement as the 100 hPa level is approached, but then swing much drier in the lowermost stratosphere and upper troposphere. This is in line with results reported with other in situ hygrometers. Reported here are some of the first comparisons with MLS v5, and the results are somewhat interesting but hardly compelling given that only two flights are presented.

E. "February 19th flight " (lines 280-308)

This section attempts to explain the differences between the October and the February flights in terms of the very different meteorological situations obtaining in the winter of 2019 and the autumn nine months later. The analysis is somewhat interesting but not particularly relevant to the rest of the paper.

TECHNICAL COMMENTS:

As noted above the text is both poorly organized and replete with grammatical as well as other kinds of errors. Both beyond the scope of journal review.
* * *

---

## Author Comment (AC1) · 19 Apr 2021

First of all, the authors thank referee#3 for his/her valuable comments and suggestions. A thorough revised writing has been conducted and further analysis conducted to end on this revised manuscript. Then, the manuscript is strongly different than the original version. The English has been revised by one of our collaborator who is a native English speaker and a specialist in hygrometry. Please to find our response to your comments below :
MAJOR COMMENTS:

Following reviewer's comments, discussion on error sources has been thoroughly revised and completed. The expended discussion can be found in the section 2.6. Uncertainties. If the original discussion about the filament has been removed, a new discussion around the filament has been added based on GRUAN consistency which has been added to the analysis. In this frame, the contribution of the filament to the analysis has brought values to the understanding of the observed discrepancies.

DETAILED COMMENTS:

-From lines 17 to 90: the introduction has been completely revised. -Line 115: the discussion about line shape effect can be found in the section 2.5 Spectroscopy. It has been extended to the troposphere. It is shown that, due to the spectra signal to noise ratio and possibly to the weakness of line parameters, even at tropospheric pressures, no high order line shapes are observed. The first author has an extended experience in the study of line shape parameters at low temperatures for atmospheric applications (e.g. OCO-2, MERLIN space mission and Pico-SDLAs: Ghysels et al., 2011, 2013a, 2013b, 2017, Delahaye, 2019). An additional figure is used to illustrate the residuals from the fitting of tropospheric spectra supporting the affirmation (figure 6, see below).

Figure 6 shows examples of 4 atmospheric spectra of the the $413 \leftarrow 414$ line of water vapor in the lower troposphere (highest pressure, where speed-dependence of line width becomes important) at 896.8, 819.1, 722.9 and 567.9 hPa. One can see that residuals using Voigt profile are flat (no sign of non-Voigt effects). For these spectra, the signal-to-noise ratio scaled from 1200 to 1600. About speed dependence of line width, residuals can be seen while SNRs are close to 10000 or above, which is not the case here.

-Line 121: the balloon is bursting once the maximum altitude is reached. Authors have corrected sentences accordingly.

-Lines 140: an expended description of the spectra fitting procedure is given in section 2.5 Spectroscopy (paragraph from line 259 to 282) and figure 7 has been added to illustrate some of the main features.

It has to be noted that the differential spectra is only used onboard as a tool to determine the peak position. This procedure is used to compensate for any spectral drift by adjusting the temperature of the laser semiconductor, therefore, it has not any involvement in the retrieval procedure. To calculate this differential spectrum, the electrical ramp is used balanced with specific gain.

- Line 149: After a careful checking, one spectrum acquisition is faster than expected, it is 8 ms instead of 20ms. In reality the measurement cycle (of a duration of 1 second) is such that: the first 200 ms are devoted to the acquisition of 5 elementary spectra and the other 800 ms are devoted to the acquisition of pressure, temperature and other technical parameters which are used for diagnostic. The pressure is averaged over half a second and the temperature is acquired within 1 millisecond during which 20 measurements are taken, filtered to remove outliers and averaged. The pressure and temperature mean values are stored on the SSD disk and will be the input of the spectra processing. As requested, we have added quantitative information on the impact of pressure and temperature measurements on the spectra processing in section 2.6. Uncertainty (line 286-319). Table 2 has been revised so it is easy to find Pico-Light water vapor total uncertainty and its vertical resolution corresponding to MLS pressure level (see new table below).

Table 2. The relative uncertainties u of measurements of temperature T and mixing ratio X made by Aura MLS and Pico-Light H2O. Also shown is the resolution $\delta(z)$ of the height z.

See table in PDF attached.

The relative uncertainty induced by errors on pressure and temperature, as well as from other parameters like the baseline, frequency axis estimations, have been calculated using synthetic spectra reproducing real atmospheric spectra in terms of baseline variability and instrumental noise and with the introduction of virtual pressure and temperature errors. The uncertainty has been obtained from the deviation of the retrieval to the real value. About the temporal resolution: Several aspects have to be considered: 1- We are limited at some point with the cycle duration. To have an autonomous instrument, the embedded software has to deal with different technical parameters and it takes some time to acquire all the necessary parameters for: 1- in real time, make sure the instrument will behave properly and 2- after flight, having essential diagnostic parameters saved so it is easy to figure out the reason in case of failure (minor or major). In fact, it takes 800 ms to manage everything in-flight. 2- For temperature, it is easy to acquire a sufficient amount of data within 8ms (here we use only 1 ms for 20 measurements) and having a temperature measurement which is improved in terms of precision. About the pressure sensor, Honeywell ppt1 has a resolution of at worst 0.1% FS (here 0.1 hPa at worst and 0.01 hPa at best). During the descent, the pressure variation over half a second is of 0.05 hPa near burst altitude up to 0.2 hPa at 800 hPa. But since the absolute pressure accuracy is of about 0.5 hPa, we consider that averaging pressure measurement over half a second won't introduce significant bias to the pressure measurement but will further improve the measurement precision.

However, if requested, it is easy to interpolate pressure and temperature at the same frequency rate as spectra.

-Line 151 : yes, measurements are taken on parachute descent. As requested we have added figure 4 illustrating the flight profile (altitude and fall speed).

-Line 162: the description about how air temperature measurements are obtained has been expended an improved (line 184-188). The temperature of concern on line 162 (old manuscript) is the temperature of the enclosure of laser diode. Indeed, the emission center frequency is sensitive to the surrounding environment temperature. To limit the emission frequency drift, the enclosure of the laser diode has to be maintained within an acceptable range of temperature and it is this correction which is mentioned

here. However, for Pico-Light, the frequency detuning is set quite large to ensure that, in case of failure in this enclosure temperature regulation, the absorption line would remain within the scan. Anyhow, the temperature correction does not have any impact on the retrieval since one spectra is recorded within 8 ms, which is too quick to integrate the frequency shift of the line (no line distorsion). Additionally, the design of the thermal enclosure is such that only a small correction of temperature is needed.

-Line 178: no remote control means that the instrument has no TM/TC onboard. All data are stored onboard a SSD card and therefore, the instrument recovery is needed to obtain the data, unlike for former Pico-SDLAs.

-Line 192: Imet-4 is flown as piggy back only as a backup in case of failure of the GNSS system onboard Pico-Light. In this case, latitude/longitude/time data from Imet-4 are used. Pressure and temperature measurements would be only used if necessary in case of failure, at the cost of accuracy. RH measurements from Imet sonde can be used to compare humidity measurement in the lower troposphere if needed but are not used here anymore. Indeed, the comparison with imet RH in the lower troposphere has been removed since it was of less importance comparing to the GRUAN consistency discussion.

-Line 193: the absolute uncertainty has been added. In this case it is 0.5 hPa and its impact in the uncertainty budget has been discussed in section 2.6. Uncertainty (line 302-306).

-Line 206: yes, the sampling rate of Honeywell PPT1 can be as high as 120 samples/sec. Pressure measurements are averaged by the sensor based on the desired averaging time (here half a second). The reasons for the selection of averaging time are detailed for the discussion about line 149 above.

-Line 206: right, I was considering the physical acquisition time. "In the previous section you mentioned that the spectra a measured only for 200 ms, followed by temperature and pressure measurement and processing. If that is the case, then the uncertainty

over 200 ms should be the same as over 1 s." To clarify, in the text we have added the following sentences: "The mixing ratio standard deviation in the stratosphere, and therefore, the precision, is of about 277 ppbv while using unitary spectra (no averaging) which corresponds to a precision of 130 ppbv for a 1 second integration time (co-addition of 5 spectra over 200 ms). " In practice, due to the large amount of measurements, it is additionally possible to filter the high resolution profile through moving windows of 20 data points without altering vertical structures (to smooth the vertical profile). Doing this allows to further virtually improve the precision.

-Line 210: this has been done, as specified in response to the comments on line 151.

Line 262: in the old version of the manuscript, the MODIS water vapor maps were only used to check whether Pico-Light and MLS were sounding the same airmass (i.e. ozone-enriched for polar air mass). In the revised analysis and manuscript we have used MERRA 2 ozone fields instead to do the same job since ozone is a dynamic tracer. The MODIS contribution has subsequently been removed.

-Section 6.1 : as specified earlier, temperature is measured using Sippican NTCs onboard during the descent of the balloon. Indeed, in the comparison which is now included in the manuscript, is based on descent measurements from the Sippican NTCs onboard Pico-Light and ascent measurements of the platinium sondes onboard RS41. Of course, sensors are coated to minimize radiation effects and radiation correction is applied, leading to mean bias of 0.12 K in the stratosphere and 0.56 K over the full altitude range. This is in agreement with comparisons between radiosondes in the frame of WMO campaigns.

-Line 296 : this paragraph has been deleted since it does not bring any value to the new results.

-Line 322 : additional details about MLS are now found in the section 4. Description of used datasets and the selection criteria is defined in section 5. Method for inter-comparison with Aura-MLS retrievals and selection criteria (lines 377-385). The

corresponding paragraph is : " The exclusion of improper MLS profiles was guided by two metrics output for each MLS profile: the "quality" and "status" criteria. The threshold and meaning of each criterion are given in the v4.2 and v5 data quality documents. The "status" criterion is a 32-bit integer containing several flag bits. The value of this criterion allows the user to know whether the profile is questionable and if so, the underlying reason. An odd value of this criterion means that the profile should not be used. The "quality" criterion acts as a threshold for scientific use. It is based on fits achieved by the Level 2 algorithms to the relevant radiances. Larger values of "quality" indicate better radiance fits and therefore more trustworthy data. The "quality" threshold for water vapor was set at 0.7. For both flights the "quality" criterion for each of the two MLS profiles used was above 1.87."

-Line 329-381 : the section 6 has been completely revised. The datasets have been reprocessed using new spectroscopy leading to updated comparison in section 6.2. Water vapor. Additionally, the analysis of the bias between MLS v4.2 and v5 have been reinforced based on GRUAN consistency criteria and MERRA 2 ozone 3-hourly reanalysis. In this section, we have shown that MLS v5 retrievals are dryer than MLS v4.2 (which then are less consistent with Pico-Light in general). While MLS v4.2 absolute values of mixing ratios are more consistent with Pico-Light (following GRUAN approach), the discrepancies of MLS v5 with Pico-Light are found more logical when considering meteorology (in relation with ozone-enriched air from polar latitude observed from the MERRA 2 products). Details about this are found from line 467 to 506.

Corrections about "levels" have been applied.

-Line 347 : right, we have corrected the mistake.

-Line 354: see reply on line 329-381.

-Line 381 - 400: the section 6.3 has been removed accordingly since it did not bring any interest anymore.

- Line 430-433 : corrections have been applied. Done. - Line 432 : Done. - Line 430 : The conclusion has been completely revised based on new results.

In general, figures have been almost completely replaced. Anyway, on the old figure 3 and now, the new figure 5 and 6, the spectra are unitary, meaning that no averaging has been performed.

Figure 1 has been slightly improved so it is easier to locate each element. The laser diode is located at the bottom of the cell and photodiode at the top to minimize the impact of ambient infrared radiation onto the atmospheric signal

The figure showing the comparison with ambient temperature from VIZ Sippican sonde and other datasets (e.g. MLS, ERA and RS41) is now figure 9 In this figure, for clarity, only one profile of MLS (v5) is shown since temperature profiles from both versions are almost identical. ERA 5 profiles donot appear on the vertical profile panel but bias is still shown in the right panels. The bias original panel has been split into 2 panels, one for each flight. The bias with RS41 is shown.

The comparison of water vapor profiles is now shown in figures 11 and 12.

Figure 11: Convolved vertical profiles from the Pico-Light H2O measurements (black diamonds) compared with MLS v4.2 (red open squares) and v5 (blue open squares) between 20 and 316 hPa on February 19, 2019. The right panel shows relative difference per pressure level between Pico-Light H2O and MLS datasets. (see figure attached)

Figure 12: Same as figure 11 but for October 16, 2019. (see figure attached)

In these figures, only low resolution profiles of Pico-Light convoluted with MLS averaging kernels are shown for clarity.

High resolution vertical profiles are visible in figure 10 (see figure attached).

Figure 10:Vertical profiles of water vapor from the descent of Pico-Light H2O on February 19, 2019 (right panel, black line) and October 16, 2019 (left panel, black line) together with error bars (grey shaded). The associated temperature profiles are shown as blue circles on each panel.

The gap around 200 hPa on October 16, 2019 comes from a bug in the electronics where data were not stored onboard. This problem has been solved afterward.

Old figures 10 and 11 have been removed. The comparison of Pico-Light with Imet sonde and RS 41 in the lower troposphere was of less importance compared to the discussion around GRUAN consistency (illustrated in figure 13 below) which has been added. Instead, we have included the following new figures :

Figure 13: Consistency between Pico-Light and MLS v4.2 (black) and v5 (red) on February 19, 2019. The full circles illustrate the absolute difference in mixing ratio between Pico-Light and MLS datasets. The limits for k=3, k =2 and k =1 are represented as black full lines for MLS v4.2 and asred full lines for MLS v5. The area under the k lines for MLS v5 are filled with different colors for a better visualization. Ozone stratospheric profiles from the OHP (Observatoire de Haute Provence) LIDAR are shown in blue dash (February 19, 2019) and dot (February 18, 2019) lines.

Figure 14 and 15 provide a support for the analysis coming from figure 13.

Figure 14: MERRA-2 ozone 3-hourly mixing ratio at 70 hPa on February 19, 2019, 3:00 UTC (left) and 9:00 UTC (right). The 9:00 UTC map represents dynamical conditions close to Pico-Light descent time and the 3:00 UTC map corresponds to the MLS case. Positions of Pico-Light and MLS mean position are shown in color circles.

Figure 15: Same as figure14 but at 100 hPa.

Please also note the supplement to this comment:
https://amt.copernicus.org/preprints/amt-2020-269/amt-2020-269-AC1-supplement.pdf

[Figure]

**Fig. 1.** Figure6 : Atmospheric unitary spectra of the 413←414 line of water vapor in the tropo-sphere (top panel) at four pressures between 567 and 900 hPa. The bottom panel shows the residuals from the fitting

[Figure]

**Fig. 2.** Figure 7 :

[Figure]

**Fig. 3.** Figure 4 : Typical flight profile under a Totex rubber balloon. Altitude profile is shown in black and descent fall speed profile is shown in blue.

[Figure]

**Fig. 4.** Figure 1

[Figure]

**Fig. 5.** Figure 11

[Figure]

**Fig. 6.** Figure 12

[Figure]

**Fig. 7.** Figure 10

[Figure]

**Fig. 8.** Figure 13

[Figure]

**Fig. 9.** Figure 14

[Figure]

**Fig. 10.** Figure 15

---

## Author Comment (AC2) · 19 Apr 2021

First of all, the authors thank referee#2 for his/her valuable comments and suggestions. A thorough revised writing has been conducted and further analysis conducted to end on this revised manuscript. Then, the manuscript is strongly different than the original version. The English has been revised by one of our collaborator who is a native English speaker and a specialist in hygrometry. Please to find our response to your comments below :

**GENERAL COMMENTS**

Referee#2 suggested that temperature measurements from Pico-Light H2O used in the comparison come from a radiosonde integrated in the instrument package. This comment has suggested that the instrument description was not as clear as it should have been. Therefore we have thoroughly improved the description in consequence. The temperature sondes are Sippican fast response thermistors. I believe this is what referee#2 meant by "radiosondes". We have to specify that water vapor mixing ratio measurements do not come from radiosonde measurements. The iMet-4 sonde onboard Pico-Light is only used for : 1- have backup localization and temporal information (e.g. balloon trajectory), 2- to compare if needed, humidity in the lower troposphere, in case of failure or questionable data.

"My first concern is that the temperature comparisons add little to the understanding of the performance of the Sippican VIZ sonde, certainly not without comparison against another in situ sonde temperature."

Reply: We have included temperature comparison since it is one of the input parameter for the spectra processing procedure. In spectroscopy, temperature of the sounded medium plays a role in the line area (having therefore a direct impact on retrieved mixing ratio) and in the line width. Indeed, we have added comparison with in-situ RS 41 sonde launched by CNES 1h15 later after Pico-Light on February 19, 2019. Differences with RS41 temperature measurements are within the uncertainty of RS41 in the lower stratosphere. Considering temperature data from ground to 20 hPa, the mean bias is similar to biases between radiosondes which have been reported in the frame of WMO intercomparison campaigns (i.e.: 0.5K). Having similar results with WMO is one check point to ensure the quality of measurements. The two temperature sondes used onboard Pico-Light, are located at each ends of the optical cell (top and bottom, see new figure 1), and are VIZ NTC thermistors from Sippican. Although supposed less accurate than platinium sondes used onboard Vaisala sondes, they are more sensitive to changes in ambient temperature and the bias with Vaisala RS41 sonde reported here

allows to be confident in temperature measurements provided by the Sippican sondes used.

"With regard to the water vapor mixing ratio measurements, the MLS v4 water vapor comparisons suggest that the Pico-Light H2O hygrometer in these two flights is comparable to other in situ instruments. The MLS v5 comparison is of some interest, but the two flights present do not present the same story in the below-100 hPa region where v4 is understood to have an instrumental bias. More significantly, it goes without saying that two flights – and two flights in significantly different meteorological settings - are a very slim basis upon which to make a judgement of the performance of an instrument measuring any atmospheric trace constituent, and water vapor with its strong vertical gradients particularly so. Thus I don't see great value in the profiles presented; what would be of considerably greater interest would be head-to-head intercomparison with a reference-quality in situ hygrometer."

Reply: Head to head comparison is scheduled in 2021-2022 from the CNES Aire-sur-l'Adour balloon facility. In this frame, we will compare in-situ measurements from FPH NOAA, FLASH-B, the micro-hygromètre (from LPC2E, CNRS, France) and our Pico-Light. In between, limiting the analysis to the cases where MLS and Pico-Light were sounding the same airmasses, the comparison with MLS v4.2 in the altitude range from 20 to 147 hPa allows an indirect connection between Pico-Light and other existing hygrometers, such as FPH NOAA or CFH. The manuscript has been revised in this direction. As an example, Yan et al., 2016, where NOAA FPH has been compared to MLS v4.2, has reported bias in in-situ water vapor measurements of 11% in average between 68 and 146 hPa. In our case, the bias in the same altitude range is of 10.8%, similar to Yan et al., 2016. The similarity in the bias obtained against the same version of MLS is an indication of the performances of Pico-Light and represents encouraging results. We agree that, in the troposphere, the strong variability in water vapor content impede any valuable conclusion. However, Pico-Light is primarily designed to probe the UTLS and lower stratosphere. For this reason, tropospheric measurements are not

of primary interest.

SPECIFIC COMMENTS:

The manuscript has been re-organized and the description of the instrument itself has been expended and detailled compared to the original manuscript. The section Pico-Light H2O has been revised in a cleaner manner. The data processing has been detailed whereas was absent in the former version of the document. Details about how are used Sippican temperature measurements are given: only the coldest temperature is used since it suggests that it is less impacted by solar radiation. Additionally, temperature corrections are applied coming from the Sippican WMO testings. As expressed earlier, the iMet-4 sonde is only used as a backup in case of failure in GPS measurements.

The pressure sensor is a Honeywell PPT1 and not PPT2 (mistake from our side). The accuracy is given at $\pm$ 0.05% for PPT1 and $\pm$ 0.0375% for PPT2. PPT1 absolute uncertainty is 0.5 hPa. A section dedicated to "uncertainty" (section 2.6) has been added in the manuscript which addresses the impact of physical parameters (environmental) : pressure and temperature, but also of other parameters to the measurement uncertainty. It has been demonstrated that the uncertainty due to combined pressure and temperature induces an error of 0.3% at maximum. The largest source of uncertainty being the spectra quality.

B. "Method for intercomparison with Aura-MLS retrievals" The high resolution profile is linearly interpolated in the pressure log space to provide a low-resolution profile in the MLS pressure log space. Then, at each MLS pressure level, is associated one value of the mixing ratio from Pico-Light which comes from the linear interpolation and therefore an error bar is estimated based on the deviation of the 100 Hz values compared to the interpolated value summed in quadrature with other sources of errors. The low-resolution profile is the input to the averaging kernels.

C. "Results and discussion - Temperature" (lines 191-224) Over 1 milliseconds, 20

measurements of temperature are performed and outliers removed to calculate the average temperature which is stored onboard. As for pressure, doing this way allows to improve the precision of the temperature measurements by a factor of $\sqrt{N}$, N being the number of data points used. This procedure is used for both temperature sondes. At the end, during the spectra processing, from the two averaged temperatures, the coldest one is chosen as physical input parameter to the fitting procedure.

D. "Results and discussion – Water vapor " (lines 225-279) This section has been thoroughly revised. Analysis using the GRUAN consistency definition is performed and MERRA 2 ozone 3-hourly reanalysis are used to help the interpretation of observed biases. This analysis helped in restricting comparison with MLS on pressure levels for which both instruments were sounding the same airmasses. This has allowed a new estimate of the biases and therefore modified our conclusions in a better perspective. Processing this way allows to demonstrate that the mean bias in the lower stratosphere is strongly similar to the one reported between the CFH (Cryogenic Frost Point hygrometer, Vömel et al., 2007) and MLS v4.2 which is even better that previously estimated. This is encouraging. About MLS v5, we have observed that MLS v5 retrieval are systematically dryer than MLS v4.2. The results reported here are the first one for this version of MLS even though only 2 profiles are compared.

E. "February 19th flight " (lines 280-308) This section has been removed. In place, a discussion around the GRUAN consistency analysis supported by MERRA-2 ozone reanalysis has been added and allows to explain the observed discrepancies on some of the pressure levels.

Please also note the supplement to this comment: https://amt.copernicus.org/preprints/amt-2020-269/amt-2020-269-AC2-supplement.pdf